# An approximate sampler for energy-based models with divergence diagnostics

**Bryan Eikema**[*]  
*University of Amsterdam*  
*b.eikema@uva.nl*

**Germán Kruzsewski**  
*NAVER Labs Europe*  
*german.kruzsewski@naverlabs.com*

**Christopher Dance**  
*NAVER Labs Europe*  
*christopher.dance@naverlabs.com*

**Hady Elsahar**[†]  
*Meta AI*  
*hadyelsahar@meta.com*

**Marc Dymetman**[†]  
*Independent researcher*  
*marc.dymetman-contractor@naverlabs.com*

**Reviewed on OpenReview:** *https://openreview.net/forum?id=VW4IrCOnOM*

## Abstract

Energy-based models (EBMs) allow flexible specifications of probability distributions. However, sampling from EBMs is non-trivial, usually requiring approximate techniques such as Markov chain Monte Carlo (MCMC). A major downside of MCMC sampling is that it is often impossible to compute the divergence of the sampling distribution from the target distribution: therefore, the quality of the samples cannot be guaranteed. Here, we introduce quasi-rejection sampling (QRS), a simple extension of rejection sampling that performs approximate sampling, but, crucially, does provide *divergence diagnostics* (in terms of $f$-divergences, such as KL divergence and total variation distance). We apply QRS to sampling from discrete EBMs over text for controlled generation. We show that we can sample from such EBMs with arbitrary precision in exchange for sampling efficiency and quantify the trade-off between the two by means of the aforementioned diagnostics.[1]

## 1 Introduction

Generating samples from a probabilistic model is a fundamental part of many machine learning tasks. Sometimes, the relation between the probabilistic model and the associated generative process is direct: for instance, in language modelling, an autoregressive model can both generate a sequence by ancestral sampling and compute its probability. However, the simplicity of sampling in such models often comes at the cost of expressivity. A family of models with much greater representational freedom is the family of *energy-based models* (EBMs) (LeCun et al., 2006). Such models map elements $x$ of the sample space to real-valued "energies" $E(x)$, or, equivalently, to non-negative scores $P(x) = e^{-E(x)}$ which can be seen as unnormalized probability distributions. However, EBMs can be difficult to sample from.

---

[*]Work done during an internship at NAVER Labs Europe.

[†]Work done at NAVER Labs Europe.

[1]To access the code for this work (scheduled for release in Jan. 2023) and for a short introductory video, please go to https://disco.europe.naverlabs.com/QRS.

In this paper, we address the problem of *sampling* from EBMs (as in "obtaining samples" and not just computing expectations, which is another popular but different problem), with a particular focus on discrete spaces of sequences over a finite vocabulary, and study applications to text generation. A popular approach to sampling from complex, unnormalized, probability distributions, such as EBMs, consists in applying Markov chain Monte Carlo (MCMC) techniques, which are guaranteed to converge to the target distribution in the limit, under mild regularity conditions (Robert and Casella, 2004). In practice, however, the length of the Markov chain is finite, in which case often only *approximate samples* are obtained. In order to know whether the samples are representative of the target distribution, ideally, one should *quantify the divergence* of the MCMC sampling distribution from the target distribution in terms of well-established metrics (e.g. total variation distance or KL divergence). Unfortunately, evaluating convergence is often challenging (Cowles and

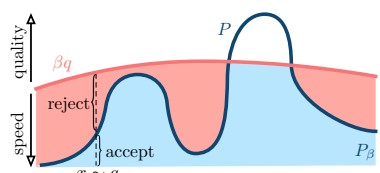

Figure 1: Quasi-rejection sampling **(QRS)** approximates a target distribution $P$ with a truncated distribution $P_\beta$ (the blue shaded area). This QRS distribution is defined in terms of a global proposal distribution $q$ and a scalar parameter $\beta$ that controls the quality of the approximation.

Carlin, 1996; Roy, 2020), especially if one makes no assumptions about the sample space. For instance, popular convergence assessments such as effective sample size (ESS) (Gamerman and Lopes, 2006) or $\hat{R}$ (Gelman and Rubin, 1992; Vehtari et al., 2021) require Euclidean structure for computing variances and correlations of *real-valued* variables (or vectors in multivariate generalizations; Vats et al., 2019; Brooks and Gelman, 1998), which are not intrinsically defined on *discrete* spaces only endowed with a probabilistic structure. For this reason, prior work on sampling from discrete sequence spaces often relies on proxy metrics (such as perplexity, diversity metrics and constraint satisfaction). Such proxies are typically insufficient to assess how representative the samples are of the target distribution, and can be misleading.

In this paper, we introduce a simple approximate sampling technique, quasi-rejection sampling (QRS), which provides explicit estimates of the divergence from the target distribution for the general class of *f-divergences*, which includes the total variation distance (TVD), forward and reverse KL, Jensen-Shannon, and $\chi^2$-divergence (Polyanskiy, 2019). This is possible because QRS associates explicit probability *scores* with samples: a property generally lacking in MCMC samplers. One may use such divergence estimates to tune the sampler, controlling the trade-off between efficiency (acceptance rate) and quality (divergence).

QRS is a relaxation of rejection sampling that obtains approximate samples from a target distribution (see Fig. 1). This requires access to a *global* proposal distribution that, ideally, produces samples close to the target distribution. Traditionally, MCMC methods work with *local* proposals (conditional distributions defined around a given point, for instance a distribution defined by performing a set of local edits), since high-quality global proposals have been hard to construct. Fortunately, with the advent of powerful neural network training techniques, this situation is rapidly changing and obtaining powerful global proposals is now possible as we show in our experiments.

We demonstrate the effectiveness of QRS on controlled text generation. Large pre-trained language models are becoming increasingly useful general purpose tools and *generation* is typically accomplished by sampling (Nadeem et al., 2020). Controlling the distribution of these language models to accommodate human preferences can be difficult, but EBMs are a promising way to achieve this (Khalifa et al., 2021). However, sampling from EBMs defined over a discrete sequence space is non-trivial, making it a challenging task to benchmark QRS. In this paper, we experiment with EBMs resulting from restricting a GPT-2 language model (Radford et al., 2019) in some way: either the model is restricted to only generate sequences containing a specific term; or the model is restricted to have certain moments, for example debiasing a distribution over biographies to consist of 50% female biographies. We explore a variety of ways to construct proposal distributions for QRS. In particular, we explore prompting a pre-trained language model, as well as training an autoregressive model to approximate the EBM (Khalifa et al., 2021). In App. B, we also experiment with a paraphrase generation task in which we use off-the-shelf machine translation models as conditional proposal distributions. Results show that we are able to approximate the target distributions to any desired level in exchange for sampling efficiency. Finally, we include experimental comparisons with both local (random-walk) and global MCMC methods, showing that QRS performs comparably or better, while providing stronger guarantees.[2]

---

[2]We will release the code for our experiments upon publication.

We summarize our main contributions as follows:

- We introduce QRS, a relaxation of rejection sampling that produces approximate samples and, in contrast to most existing approximate sampling techniques, provides estimates of sampling quality in the form of $f$-divergences.

- We prove two theorems about QRS: one provides an upper bound on the TVD between the target and QRS distributions; the second shows the $f$-divergence is a monotonic function of QRS's tuning parameter $\beta$. We also exploit the data-processing inequality to at least provide a lower bound on the divergence of MCMC samplers and compare it to the precise divergence estimate for QRS.

- We show in experiments on text generation how QRS is able to approximate the target EBMs to any desired level of accuracy in exchange for sampling efficiency.

- We show how to construct efficient global proposal distributions for text generation making use of recent advances in large language models: prompting, training objectives for approximating EBMs, and the widespread availability of pre-trained sequence models.

- We compare QRS with both local and global variants of Metropolis-Hastings, and show that QRS outperforms local variants, while performing on par with global variants, according to proxy metrics. We show, however, how proxy metrics can fail to reliably estimate the divergence from the target EBM, a quantity typically not available to MCMC.

## 2 Formal Approach

Our general problem is the following. We consider a discrete (i.e. countable) sample space $X$. We are given a nonnegative real function — such as an EBM — $P(x)$ over $X$, such that the partition function $Z \doteq \sum_{x \in X} P(x)$ is strictly positive and finite. We can then associate with $P$ a normalized probability distribution $p(x) \doteq P(x)/Z$. Our goal is to define a "sampler" $\omega$, that is a generator of elements from $X$, such that $\omega$ produces a sample $x$ with a probability $\omega(x)$ as close as possible to our target $p(x)$, in terms of distance measures such as KL divergence $D_{\mathrm{KL}}(p, \omega)$ and total variation distance $\mathrm{TVD}(p, \omega)$, and more generally the large family of $f$-divergences (Liese and Vajda, 2006), which we discuss later. To help us achieve this goal, we assume that we have at our disposal a *global proposal distribution* $q(x)$ such that i) we can effectively compute $q(x)$ (i.e. *score* $x$) for any $x \in X$, ii) we can efficiently generate samples from $q$, and iii)$p$ is absolutely continuous relative to $q$ (denoted by $p \ll q$), in other words the support of $q$ includes the support of $p$, i.e. $p(x) > 0 \implies q(x) > 0$.

### 2.1 Quasi-rejection sampling (QRS)

Our proposed sampling method, QRS, is given as Algorithm 1.

---
**Algorithm 1** QRS
---
1: **Require:** Target $P$, proposal $q$, parameter $\beta$, number of required samples $N$ $\qquad\qquad$ $\{0 < \beta < \infty\}$
2: $n \leftarrow 0$
3: **while** $n < N$ **do**
4: $\quad x \sim q$
5: $\quad r_x \leftarrow \min\left(1, P(x)/(\beta q(x))\right)$ $\qquad\qquad\qquad\qquad\qquad\qquad\qquad\qquad$ {Acceptance prob.}
6: $\quad u \sim U_{[0,1]}$ $\qquad\qquad\qquad\qquad\qquad\qquad\qquad\qquad$ $\{U_{[0,1]} :$ unif. dist. over $[0,1]\}$
7: $\quad$ **if** $u \leq r_x$ **then**
8: $\quad\quad$ output $x$
9: $\quad\quad n \leftarrow n + 1$
10: $\quad$ **end if**
11: **end while**
---

In addition to $P$ and $q$, QRS requires the input of a finite positive number $\beta$. For a given $\beta$, the QRS sampler, which we will denote by $p_\beta$, produces an independent and identically distributed (i.i.d.) sequence of

values $x$ (line 8), with a probability mass function that we denote by $p_\beta(x)$.[3] If $\beta$ is a global upper bound on $P(x)/q(x)$, then the behaviour of the QRS algorithm is identical of that of the classical rejection sampling (RS) algorithm (von Neumann, 1963). *However*, QRS does not require $\beta$ to be an upper bound, and the acceptance probability $r_x$ in line 5 is an extension of that used in RS to situations where $P(x) > \beta q(x)$. In such situations, the sample $x$ is always accepted at line 7.

QRS has crucial practical advantages over RS. It is well known that for rejection sampling, with $\beta$ a finite global upper bound, we have $p_\beta = p$: in other words, rejection sampling is a *perfect* sampler for $p$ (Robert and Casella, 2004). This is of course a major advantage, however it comes with serious theoretical and practical limitations: there may not exist such a finite upper bound, and even if one exists, its value may not be known. Furthermore, even if such a bound could be found, the resulting sampler could be extremely inefficient: the "acceptance rate" of rejection sampling is proportional to $1/\beta$, which can be very small. By relaxing the requirement that $\beta$ is a global upper bound, QRS sacrifices the identity between $p_\beta$ and $p$. However, QRS becomes much more broadly applicable, and crucially, allows an explicit trade-off between the sampling *efficiency* of $p_\beta$ and its *quality*.

## 2.2 Explicit $f$-divergence diagnostics for QRS

Let $f : (0, \infty) \to \mathbb{R}$ be a convex function such that $f(1) = 0$, let $f(0) \doteq \lim_{t\downarrow 0} f(t)$, and let $p_1$ and $p_2$ be probability distributions over a discrete sample space $X$. The *$f$-divergence of $p_1$ from $p_2$* is defined as

$$D_f(p_1, p_2) \doteq \mathbb{E}_{x\sim p_2} f\left(\frac{p_1(x)}{p_2(x)}\right) + f'(\infty)\ p_1(p_2 = 0), \tag{1}$$

where $p_1(p_2 = 0)$ denotes the $p_1$-mass of the set $\{x \in X : p_2(x) = 0\}$, and where $f'(\infty) \doteq \lim_{t\downarrow 0} t\ f(1/t)$. Here the convention is that if $p_1(p_2 = 0) = 0$, the product $f'(\infty)\ p_1(p_2 = 0)$ is taken to be equal to 0, whatever the (possibly infinite) value of $f'(\infty)$ (Polyanskiy, 2019).[4]

Unless mentioned otherwise, we will always assume in the rest of the paper that $p_1 \ll p_2$, in other words that $p_1(p_2 = 0) = 0$, so that the previous definition simplifies to

$$D_f(p_1, p_2) \doteq \mathbb{E}_{x\sim p_2} f\left(\frac{p_1(x)}{p_2(x)}\right). \tag{2}$$

The family of $f$-divergences enjoys a number of remarkable properties,[5] and contains many standard methods for measuring differences between probability distributions (Polyanskiy, 2019; Liese and Vajda, 2006). In our experiments, we mainly work with the total variation distance $\text{TVD}(p_1, p_2) \doteq \sum_{x\in X} |p_1(x) - p_2(x)|/2$, obtained with $f(x) = |1 - x|/2$, and KL divergence $D_{\text{KL}}(p_1, p_2) \doteq \sum_{x\in X} p_1(x) \log \frac{p_1(x)}{p_2(x)}$, which has $f(x) = x \log x$.

**Explicit form for $p_\beta$ and $f$-divergence estimation** Let $P_\beta(x) \doteq \min(P(x), \beta q(x))$, and let $Z_\beta \doteq \sum_{x\in X} P_\beta(x)$ be the associated partition function. Also, define the *acceptance rate* $\text{AR}_\beta$ of the QRS sampler $p_\beta$ as the proportion of samples from $q$, in line 4 of the algorithm, that are accepted on line 7, a proportion that provides a measure of the efficiency of the algorithm. We then have the following properties, proved in App. D:

$$p_\beta(x) = \min(P(x), \beta q(x))/Z_\beta = P_\beta(x)/Z_\beta, \tag{3}$$

$$\text{AR}_\beta = \mathbb{E}_{x\sim q} \min(1, P(x)/(\beta q(x))) = Z_\beta/\beta. \tag{4}$$

Eq. 3 provides an explicit form for $p_\beta$ as the normalized distribution associated with $P_\beta$, while Eq. 4 shows that the acceptance rate is a nonincreasing function of parameter $\beta$.

---

[3]This involves some abuse of notation, as $p_\beta$ refers to both a sampler and a distribution that "scores" elements of $X$.

[4]Both $f(0) \doteq \lim_{t\downarrow 0} f(t)$ and $f'(\infty) \doteq \lim_{t\downarrow 0} t\ f(1/t)$ can be shown to exist for a convex function $f$ with domain $(0, \infty)$, if we allow these limits to take the value $+\infty$ (Hiriart-Urruty and Lemaréchal, 1996).

[5]In particular $D_f(p_1, p_2) \geq 0$, and, for $f$ strictly convex at $t = 1$, $D_f(p_1, p_2) = 0$ iff $p_1 = p_2$. Also, we have $D_f(p_1, p_2) = D_{\tilde{f}}(p_2, p_1)$, where $\tilde{f}(t) \doteq t\ f(1/t)$ is another convex function with $\tilde{f}(1) = 0$ known as the *perspective* of $f$ (Remark 7.2 Polyanskiy, 2019).

This explicit form given in Eq. 3 enables us to directly compute empirical estimates of the $f$-divergence of the target from $p_\beta$. It is easy to see that, under our assumption at the beginning of the section that $p \ll q$, we have both $p \ll p_\beta$ and $p_\beta \ll p$ and in particular we can write

$$D_f(p, p_\beta) = \mathbb{E}_{x \sim p_\beta} f\left(\frac{p(x)}{p_\beta(x)}\right). \tag{5}$$

Crucially, to estimate this quantity given a collection of samples from $p_\beta$ we need to compute (or at least approximate) the two values $p(x)$ and $p_\beta(x)$ for any given $x$. And this is something that we *can* do with QRS thanks to the explicit form of $p_\beta$ of Eq. 3 and given that we can estimate the partition functions $Z$ and $Z_\beta$ — see Eqs. 6 and 7 below.

The contrast here with a typical Markov chain based sampler $\omega$ is striking: it is usually unfeasible to estimate the probability $\omega(x)$ for a given $x$ (even one sampled from $\omega$): to do so, one might estimate the chain's transition matrix or kernel, and repeatedly multiply by it, but this matrix is usually huge, or even infinite. In other words, unlike QRS, these samplers are not "scorers", making it impractical to estimate $f$-divergences $D_f(p, \omega)$ as in Eq. 5.

**Divergence estimates via importance sampling** Computing divergences as described above requires obtaining a different set of samples for every value of $\beta$. More conveniently, one can use a *single* collection $\{x_1, \ldots, x_N\}$ of i.i.d. draws from $q$ and use importance sampling (IS) (Owen, 2013) to compute such quantities. Let $h$ be a real-valued function on $X$, such that $\sum_{x \in X} h(x)$ is well defined (it may be infinite) and let $q(x) = 0 \Rightarrow h(x) = 0$. Then we can then rewrite $\sum_{x \in X} h(x) = \sum_{x \in X : q(x) > 0} q(x) \frac{h(x)}{q(x)} = \mathbb{E}_{x \sim q} \frac{h(x)}{q(x)} \simeq N^{-1} \sum_{i=1}^{N} \frac{h(x_i)}{q(x_i)}$, to compute:

$$Z \simeq N^{-1} \sum_{i=1}^{N} \frac{P(x_i)}{q(x_i)}, \qquad Z_\beta \simeq N^{-1} \sum_{i=1}^{N} \frac{P_\beta(x_i)}{q(x_i)}, \tag{6}$$

$$D_f(p, p_\beta) \simeq N^{-1} \sum_{i=1}^{N} \frac{P_\beta(x_i)}{Z_\beta q(x_i)} f\left(\frac{Z_\beta \, P(x_i)}{Z \, P_\beta(x_i)}\right). \tag{7}$$

In Eq. 7, we exploit the fact that $P_\beta(x) = \min(P(x), \beta q(x))$ is known explicitly.

More details are provided in App. E about how Eq. 7 specializes in the case of TVD and KL divergences, the measures we use in our experiments. There, we also describe the IS formulas we use for estimating the acceptance rate $\mathrm{AR}_\beta$, the expected value $\mathbb{E}_{x \sim p_\beta} h(x)$ of a function $h$ expressing a constraint, as well as the probability $p(\bar{A}_\beta)$ (see next paragraph).

**Remark about partition function estimates** As the sample mean is an unbiased estimator of the mean, the estimates in Eq. 6 are unbiased. These estimates converge (almost surely) to $Z$ and $Z_\beta$ for $N \to \infty$, a consequence of the strong law of large numbers (Tao, 2008), whether or not the random variables (RVs) $\frac{P(x)}{q(x)}$ and $\frac{P_\beta(x)}{q(x)}$ have finite variances. However, in order to provide guarantees about the accuracy of these estimates — e.g. in terms of confidence bounds — one would need to estimate these variances, or their $(1 + \alpha)$-moment for some $\alpha > 0$ as discussed by Lugosi and Mendelson (2019, p. 8). A *practical* approach consists in providing *empirical* variance estimates based on the same $N$ samples, and this is what we will do in several experiments (see Sec. 3), comforting us about the *practical* accuracy of our estimates of $Z$ and $Z_\beta$. However, in theory, the empirical variance estimates could themselves be wrong, resulting in an estimation circularity. The only way to avoid this circularity, that we are aware of, consists in cases where the RVs $\frac{P(x)}{q(x)}$ and $\frac{P_\beta(x)}{q(x)}$ can be bounded *a priori*, in which cases the variances of these RVs can also be formally bounded using Popoviciu's inequality (Popoviciu, 1935) or, more aggressively, confidence intervals produced through Hoeffding's inequality (Hoeffding, 1994), so that confidence bounds for $Z$ and $Z_\beta$ can be obtained. Interestingly, the random variable $\frac{P_\beta(x)}{q(x)} = \min(P(x)/q(x), \beta) \leq \beta$ appearing in $Z_\beta$ is bounded *by construction*, so that one can always provide formal guarantees about the $Z_\beta$ estimate. By contrast, in the case of $Z$, while one can often find proposals $q$ for which a bound on $\frac{P(x)}{q(x)}$ is known, such proposals can be

unacceptably inefficient, while other proposals, closer to the target, are much more efficient in practice, but eschew strict formal guarantees. We provide details and compute such provable bounds on $Z$ for some of the EBMs that we consider in our experiments in App. I.[6]

### 2.3 Two theorems about QRS: bounding and monotonicity

**TVD bounds on QRS** We first provide a simple upper bound on the TVD between the sampling and target distributions.

Let $\bar{A}_\beta \doteq \{x \in X : P(x)/q(x) > \beta\}$ be the set of "violators" of the $\beta$ bound. It is intuitive that the "fewer" violators there are, the closer $p_\beta$ is to the target $p$. The following theorem, proven in App. D, makes this statement precise, in terms of the actual $p$-mass of the violators:

**Theorem 2.1.** $\mathrm{TVD}(p, p_\beta) \leq p(\bar{A}_\beta)$.

As a corollary of this, observing that $\lim_{\beta \to \infty} p(\bar{A}_\beta) \to 0$, one sees that $p_\beta$ converges to $p$ for $\beta \to \infty$.[7]

**$f$-divergence monotonicity of QRS** While the previous upper bound is specific to TVD, QRS enjoys an attractive *monotonicity* property for all $f$-divergences. We provide two proofs of this result. The first proof, given in App. D.3, relies on fundamental "majorization" properties of convex functions, does not involve the differentiation of the divergence, and provides some graphical intuition in terms of gaps associated with Jensen's inequality. The second proof,[8] given in App. D.4, replaces a large part of the first proof by a much shorter calculus-based derivation that exploits an extension of the fundamental theorem of calculus not requiring differentiability, but only absolute continuity.

**Theorem 2.2.** *Let* $0 < \beta < \beta' < \infty$. *Then* $D_f(p, p_{\beta'}) \leq D_f(p, p_\beta)$.

Theorem 2.2 guarantees that increasing the value of parameter $\beta$ never increases the $f$-divergence of the target distribution from the QRS distribution. Moreover, in the light of footnote 5, increasing $\beta$ never increases the divergence of the QRS distribution from the target distribution.

## 3 Experiments

### 3.1 Approximating a Poisson distribution and the quality-efficiency trade-off

To demonstrate the usage of QRS we start with a toy setting using two Poissons. The goal is to sample from a target Poisson distribution $p$ with rate $\lambda_p = 11$ using samples from a proposal Poisson distribution $q$ with rate $\lambda_q = 10$. Rejection sampling is *not* possible in this setting as the ratio $p(x)/q(x) = (e^{-11} 11^x/x!)/(e^{-10} 10^x/x!) = e^{-1} 1.1^x$ can take arbitrarily large values when $x$ increases, i.e. the ratio is unbounded. However, it is possible to use QRS here.

We perform ten independent experiments in which we sample $10^4$ elements from $q$ that we use to compute the quality of the approximation by estimating: $\mathrm{TVD}(p, p_\beta)$ and its upper bound $p(\bar{A}_\beta)$. (See App. C.1 for additional results on $D_{\mathrm{KL}}(p, p_\beta)$.) In all cases, we use $\beta$ values in the interval $[0.5, 3.5]$. Furthermore, we compute the sampler's efficiency by estimating the acceptance rate (AR) for each value of $\beta$ (Eq. 4). As described previously and detailed in App. E, we compute estimates for all these metrics using importance sampling. Results are displayed in Fig. 2. As shown, using higher values of $\beta$ improves the TVD, even though

---

[6] In principle, it is possible to construct examples where $Z = \mathbb{E}_{x \sim q} \frac{P(x)}{q(x)}$ is finite while the variance $\mathrm{Var}_{x \sim q} \frac{P(x)}{q(x)}$ is infinite, see App. I. In practice, for text generation, we have not observed such situations. Note that all the EBMs of Sec. 3.2 are of the form $P(x) = a(x) b(x)$, with $a$ an autoregressive model and $b$ upper-bounded; therefore if one takes $q = a$, then $P(x)/q(x)$ is bounded so the variance is bounded, although the proposal $a$ can be inefficient.

[7] One reviewer asked whether one could provide guarantees about the asymptotic speed of convergence to $p$. When the support of $P$ is a finite set, the answer is easy and positive: there exists a value $\beta_0$ such that $\beta > \beta_0$ implies that $p_\beta$ is identical to $p$, by simply taking $\beta_0 = \max\{P(x)/q(x) : x \in \mathrm{Supp}(P)\}$. However, when the support of $P$ is an infinite set, we prove in App. K the following negative result (Theorem K.1): for an arbitrarily slowly decreasing function $g(\beta)$ that converges to 0, there exists some proposal $q$ such that $\mathrm{TVD}(p, p_\beta)$ converges *even more slowly* than $g(\beta)$. In other words, if one plays "adversarially", it is always possible to design an arbitrarily bad proposal in terms of asymptotic convergence rate.

[8] We thank one of the reviewers for suggesting this proof.

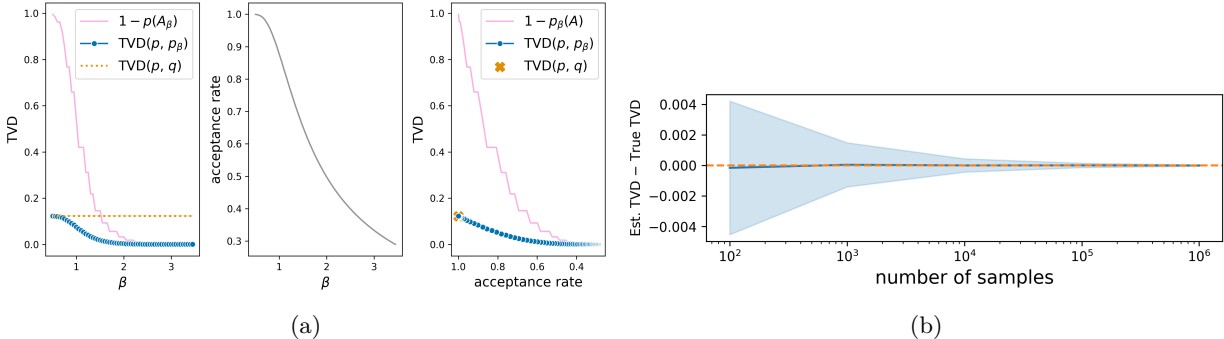

(a)                                   (b)

Figure 2: (a) Estimation of sampling quality as $\text{TVD}(p, p_\beta)$, efficiency (acceptance rate), and the trade-off between them for a QRS sampler when using a proposal $q = \text{Poisson}(\lambda = 10)$ to approximate $p = \text{Poisson}(\lambda = 11)$ computed in 10 independent experiments over $10^4$ samples. (b) Differences between estimated $\text{TVD}(p, p_\beta)$ and their true values for $\beta = 2$, computed 1000 times for each different number of samples. The blue line is the mean and shaded areas represent one standard deviation.

this comes at the cost of lower acceptance rate. In particular, with $\beta = 3.5$, the TVD is tiny ($\approx 10^{-4}$), yet the acceptance rate is moderate (0.3, i.e. 30% of proposal samples are accepted). Notably, we can ease the visualization of the trade-off between quality and efficiency by reparametrizing the divergence metrics in terms of the acceptance rate (last panel of Fig. 2a). A procedure to map acceptance rates to a corresponding $\beta$ is outlined in App. F. We use this concise reparameterized representation to plot subsequent results.

**Empirical estimates of the divergence diagnostics** In the previous and following experiments, we use a sufficiently large sample size to obtain accurate estimates of the divergence diagnostics. However, it is reasonable to wonder about the bias and variance of these estimators for smaller sample sizes. While it is not possible to provide a definite answer for all EBMs (cf. Sec. 2.2), we can investigate this question by exploiting the fact that we can compute with great precision $D_f(p, p_\beta)$ when both $p$ and $q$ are Poisson distributions (see App. H for a derivation). We compare this approximation to the true value with the estimators proposed in Sec. 2 using sample sizes $n \in \{10^2, \dots, 10^6\}$ and repeating the process 1000 times. Results for TVD are shown on Fig. 2b (see App. C.1 for results on KL). As it can be seen, there is some slight variance when only 100 samples are used for the estimation, and it quickly improves as more samples are used. In contrast, the estimation bias is tiny.

## 3.2 Generation with distributional control

The following experiments focus on the task of generation with distributional control, introduced by Khalifa et al. (2021), a task that requires sampling from an EBM over sequences of discrete tokens, making it an ideal test bed for QRS. Given a language model $a(x)$, the goal of this task is to sample from a model $p(x)$ that, on the one hand, constrains the moments of a vector of $n$ pre-defined features $\phi(x)$ to match some desired value $\bar{\mu}$ (i.e. $\mathbb{E}_{x \sim p}\phi(x) = \bar{\mu}$), while on the other hand minimizing $D_{\text{KL}}(p, a)$, a generalized version (Csiszar, 1975; Kullback and Khairat, 1966) of the maximum entropy approach (Jaynes, 1957; Rosenfeld, 1996). For example, one might want to debias a language model trained on a corpus of biographies to produce biographies only of scientists, 50% of which should be female. Then $\phi_1(x)$ and $\phi_2(x)$ would be binary classifiers assessing whether a sentence speaks about a scientist or a female respectively, and the desired moments would be set to $\bar{\mu} = [1, 0.5]$.

The authors show that $p$ can be expressed as an unnormalized EBM $P(x) = a(x)b(x)$, and describe two choices of $b(x)$. On the one hand, they consider *pointwise* constraints, where $\bar{\mu} \in \{0, 1\}^n$. For instance, if there is a single binary feature for which we would like that $\forall x : \phi(x) = 1$, then $b$ takes the form $b(x) = \phi(x)$. Otherwise, in the case of *distributional* constraints in which $\bar{\mu} \in \mathbb{R}^n$, they show that there is a vector $\lambda \in \mathbb{R}^n$ such that $b(x) = \exp(\lambda \cdot \phi(x))$ and $p(x) \propto a(x)b(x)$ fulfills the requirements of moment matching and minimal

| prompt name | prompt |
|---|---|
| simple | Wikileaks. |
| multiple | Wikileaks, Wikileaks, Wikileaks. |
| knowledge | Here is what I know about Wikileaks: |
| jeopardy | This medium was founded by Julian Assange in 2006. |
| news | Here are the latest developments on Wikileaks: |

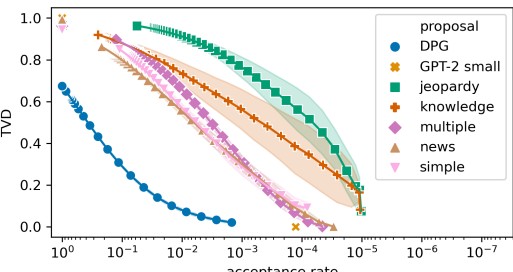

Figure 3: Comparing GPT-2, GPT-2 conditioned on various prompts, and a fine-tuned model (DPG) as proposals for generating sequences containing "Wikileaks". Disconnected points in the upper-left corner indicate $\text{TVD}(p, q)$ for each proposal, while the curves show $\text{TVD}(p, p_\beta)$ as a function of the acceptance rate. Standard deviation bootstrap estimates are shown as shaded regions for every proposal.

KL divergence from the original model. The vector $\lambda$ is found using self-normalized importance sampling (Owen, 2013; Parshakova et al., 2019a) and stochastic optimization.

### 3.2.1 Proposals for a pointwise constraint

We first experiment with constraining GPT-2 small (Radford et al., 2019) using one of the pointwise constraints ($\bar{\mu} = 1.0$) proposed in Khalifa et al. (2021), namely, $b(x) = \mathbb{1}[x$ contains "Wikileaks"]. In order to apply QRS we need to find a suitable proposal distribution. A possible candidate is GPT-2 small itself. An advantage of this proposal is that we can use pure rejection sampling with an upper-bound $\beta = 1$ to obtain exact samples from the EBM. This is because we can upper bound the ratio $P(x)/q(x) = a(x)b(x)/a(x) = b(x) \leq 1$. In fact, for $b(x) \in \{0, 1\}$ this process reduces to "naively" filtering out all samples for which $b(x) = 0$. However, a serious disadvantage is that the acceptance rate will be given by the natural frequency of the constraint. Using QRS, we can employ proposal distributions leading to better efficiency at a small cost in quality of approximation to $p$. We explore two such options:

1. First, we make use of the model proposed by Khalifa et al. (2021), which consists of a fine-tuned autoregressive model obtained by applying the *distributional policy gradient (DPG)* algorithm (Parshakova et al., 2019b) to approximate the target EBM in a generic way. While this model is considerably better at satisfying the desired constraints, it does not match the desired distribution perfectly.

2. Second, in the spirit of "in-context learning" (Brown et al., 2020), we propose to *condition $a(x)$ on a prompt* with the aim of increasing the constraint satisfaction rate in the resulting conditional distributions. In contrast to the previous approach, this does not require the training of a new model, even though it does require the manual selection of promising prompts. We experiment with five such prompts, which we present in Fig. 3.

Fig. 3 shows the $\text{TVD}(p, \cdot)$ as a function of acceptance rate for different samplers. In this and the following experiments, we chose a range of $\beta$ values that yields acceptance rates in the range $10^0$–$10^{-5}$ (cf. Table 2 in App. A), using Algorithm 2 in App. F for this purpose. We first show the $\text{TVD}(p, q)$ for each proposal $q$, at an acceptance rate of 1, before applying QRS (in the upper left corner). Then, we plot $\text{TVD}(p, p_\beta)$ as a function of acceptance rate for each proposal distribution. We compute IS estimates of the TVD on 1M samples from each proposal distribution. Variance is estimated using the bootstrap estimator (Wasserman, 2010). Note that all samples obtained from QRS satisfy the constraints perfectly, as sequences that do not satisfy the constraint are always rejected, and for this reason the curves start with different acceptance rates. As expected, using GPT-2 small results in zero TVD, but comes at the cost of low efficiency, with an acceptance rate around $10^{-4}$. Using prompting, we improve the constraint satisfaction of the resulting proposal distributions and trade-off approximation quality for greater efficiency using QRS: For instance, if a TVD of 0.3 can be tolerated, then some of the prompt proposals provide a 10-fold higher acceptance

Chandra Pradha Towni (born February 11, 1965) is a social scientist, activist, poet, and author living in Portugal. She is. . .

Enrella Carrière is a Canadian writer, translator, and philosopher specializing in the history of show business. She has covered topics such as the direction and psychology of television and the evolution of human. . .

Albert Fahn (born 1970) is an American scientist who focuses on algorithms for generating biomechanical data. Methods to generate and construct biomechanical data. . .

Wyndham Radnor (born 1946) is a British historian and criminologist specialising in the subject of labour law. He has written extensively on. . .

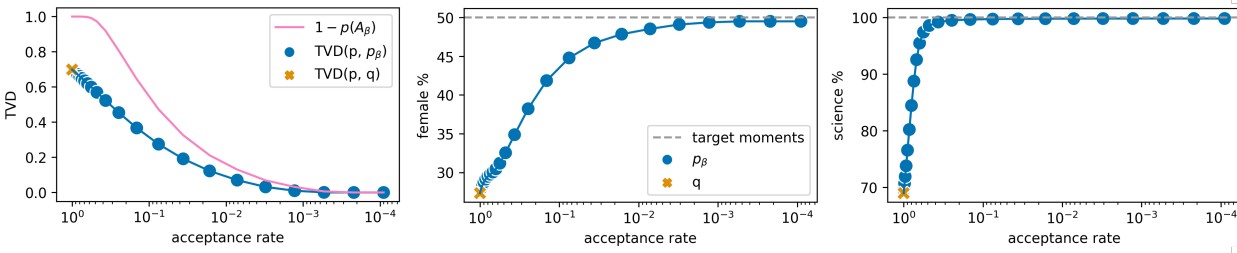

Figure 4: Estimation of the divergence from the EBM (TVD), and moments of features *female* and *science* in sampling debiased GPT-2 biographies talking about scientists. Variance is negligible as shown in Table 10 in App. I. We also show samples from running the QRS sampler at an acceptance rate of $10^{-3}$. Samples are cut off at 40 (subword) tokens and are manually chosen to show two male and two female biographies, for constraint satisfaction (moment matching) results refer to the graph. We color words that fire our female or science features.

rate with respect to the base GPT-2 model with no further training. Some prompts work notably better than others and we do not exclude the possibility of there existing prompts that perform even better than the ones we tested; we leave a more extensive exploration of prompting to create proposal distributions for future work. The autoregressive policy obtained from the DPG algorithm is the best proposal distribution we tested. Notably, it allows one to obtain low TVDs at higher acceptance rates than is possible by naively filtering samples from the base language model. For example, we can obtain a TVD of 0.1 at $100\times$ the acceptance rate (see Table 10 in App. I for more numerical results).

### 3.2.2 Distributional constraints

We now turn to the task, also introduced by Khalifa et al. (2021), of generating biographies of scientists while debiasing the gender distribution to contain female scientists 50% of the time. For this we make use of GPT-2 Biographies ($a(x)$), a language model fine-tuned on Wikipedia biographies[9] and follow the same setup as the authors to define the binary classifiers identifying sequences talking about scientists or females[10] and infer an EBM that matches the distributional constraints with minimal deviation from the original model. The frequency with which the model $a(\cdot)$ generates scientist biographies is 1.8%, female biographies 7.5%, and the frequency with which it generates female scientist biographies is only 0.14%. As proposal distribution, we use the DPG model that Khalifa et al. (2021) trained to approximate the EBM, which reaches a constraint satisfaction of 69.0% scientist, 27.3% female and 19.6% female scientist biographies. (See Table 7 in their paper for examples generated by this proposal.)

As before, we obtain 1M samples from the proposal distribution to compute importance sampling estimates (detailed in App. E) of $\text{TVD}(p, p_\beta)$, acceptance rate ($\text{AR}_\beta$), and the moments of the features that we wish to control. We show all metric curves as a function of acceptance rate of the QRS algorithm as well as some generation examples in Fig. 4.

We find that both $\text{TVD}(p, p_\beta)$ and the upper-bound on $\text{TVD}(p, p_\beta)$ steadily converge to zero as the acceptance rate decreases, meaning that we can perfectly match the target EBM in exchange for sampling efficiency. As a result, at an acceptance rate of $\text{AR}_\beta = 10^{-3}$ we nearly perfectly debias the original language model

---

[9]https://huggingface.co/mkhalifa/gpt2-biographies
[10]Gender is estimated by the ratio of female to male pronoun counts, scientists are identified by the mention of at least one of multiple words associated with the profession.

while exclusively generating biographies about scientists (49.5% female and 99.8% scientist biographies). We show some example generations at $\mathrm{AR}_\beta = 10^{-3}$ chosen manually to illustrate two male and two female biographies. Notably, we also achieve a good level of constraint satisfaction (48.4% female and 99.9% scientist biographies) and a TVD of 0.1 at $\mathrm{AR}_\beta = 10^{-2}$. This is a considerable improvement in quality with respect to the proposal distribution (with a TVD of 0.7), and in acceptance rate relative to directly rejecting from GPT-2 Biographies (which would result in $\mathrm{AR} \le 2 \times 0.14\% = 2.8 \times 10^{-3}$). Two more pointwise and two more distributional constraints including additional metrics such as $D_{\mathrm{KL}}(p, p_\beta)$ and $D_{\mathrm{KL}}(p_\beta, a)$ are shown in App. A with similar results.

### 3.2.3 Comparison with MCMC techniques

We now compare QRS with MCMC samplers, for which we focus on the EBM for a *pointwise* constraint restricting GPT-2 to only generating sequences containing "amazing".

**Baselines**  We use the popular Metropolis-Hastings (MH) algorithm (Metropolis et al., 1953; Hastings, 1970). This algorithm works by constructing a Markov chain of dependent samples, in which the next state of the chain is equal to a newly proposed sample with a certain acceptance probability, and otherwise the next state is the same as the current state. When the length $n$ of the chain tends to infinity it can be proven (Robert and Casella, 2004, Theorem 7.4), see App. J, that the average of a statistic on the elements of the chain converges to its expected value under the target distribution and, more importantly when focussing on sampling as we do, that the distribution of the $n^{\mathrm{th}}$ element of the chain converges (in total variation) to the target distribution. In practice, the chain is of finite length and only approximate samples are obtained.

Common practices are to discard the first few samples of the chain to reduce the effects of poor starting conditions (which is known as burn-in), and to only keep every $t^{\mathrm{th}}$ sample to reduce autocorrelations (which is known as thinning). We use both these heuristics in our experiments. We set a burn-in period of 1,000 steps and only keep every $1,000^{\mathrm{th}}$ sample to attain an acceptance rate of $10^{-3}$. Note that we chose not to include the burn-in period to compute the acceptance rate of MCMC samplers, as this period is constant and does not grow with sample size. We also experiment with a reset variant (-R) of the MH samplers that does away with autocorrelations among samples altogether (i.e. produces i.i.d. samples like QRS) by, instead of using thinning, resetting the chain after 1,000 steps and only retaining the last sample of the chain (see Robert and Casella 2004, Theorem 7.4 (ii)). This variant does not make use of a burn-in period.

We experiment with two proposal distributions for use in the MH samplers: i) the global proposal distribution used in QRS, i.e. DPG, for independent MH (IMH) (Robert and Casella, 2004, Sec. 7.4) and ii) a local proposal distribution that makes local edits to an evolving sample in the chain for random-walk MH (RWMH). As far as we are aware, IMH is not commonly employed in the literature due to global proposals classically being difficult to come by. We stress that with the advances in neural network training techniques such as DPG, global proposals are more accessible than ever and we therefore include IMH as a strong baseline in our experiments. Our design of the local proposal is inspired by uses of MCMC in controlled text generation, in particular by Miao et al. (2019) and Goyal et al. (2021). The local proposal randomly performs either an insert, delete or replace operation on the token level, where insert and replace operations are performed by sampling from BERT (Devlin et al., 2019). Locations of insertions and deletions are chosen uniformly at random. To inform the local proposal distribution about the target distribution, we implement the insert and replace operations by randomly mask-filling using BERT (99% of the time) and filling with "amazing" (1% of the time). We initialize the chain using a sample from the global proposal distribution.[11]

**Metrics**  As no $f$-divergence estimates are available for the MCMC samplers, we instead resort to some proxy metrics specific to controlled text generation to measure the sample quality, extracted from $10^4$ samples over ten independent experiments. In particular we measure constraint satisfaction (% amazing), perplexity (PPL) of our samples under GPT-2, diversity across samples with Self-BLEU-5 (Zhu et al., 2018) and percentage of unique samples (% Uniq), and finally, diversity within samples as given by the percentage of

---

[11]Note that localized MCMC methods can still take advantage of good starting points, especially the reset variants. We therefore also provide our variants of RWMH with a global proposal distribution for obtaining good starting points, the same global proposal as used for QRS.

| Method | %Amazing | PPL↓ | Self-BLEU-5↓ | %Uniq↑ | Dist-2↑ | TVD↓* |
|--------|----------|------|--------------|--------|---------|-------|
| proposal | $62.9 \pm 0.4$ | $61.7 \pm 0.3$ | $85.8 \pm 0.1$ | $100 \pm 0.0$ | $96.1 \pm 0.0$ | $0.67$ |
| RWMH | $100 \pm 0.0$ | - | $99.8 \pm 0.2$ | $32.0 \pm 33.7$ | $83.8 \pm 17$ | Unk. |
| RWMH-R | $100 \pm 0.1$ | $58.7 \pm 3.3$ | $87.6 \pm 0.4$ | $100 \pm 0.0$ | $92.0 \pm 0.3$ | Unk. $(\geq 0.28 \pm 0.02)$ |
| IMH | $100 \pm 0.0$ | - | $86.9 \pm 0.3$ | $98.7 \pm 0.5$ | $96.3 \pm 0.1$ | Unk. |
| IMH-R | $100 \pm 0.0$ | $63.4 \pm 1.5$ | $86.7 \pm 0.1$ | $100 \pm 0.0$ | $96.3 \pm 0.1$ | Unk. $(\geq 0.01 \pm 0.01)$ |
| QRS | $100 \pm 0.0$ | $62.8 \pm 1.6$ | $86.6 \pm 0.2$ | $100 \pm 0.0$ | $96.3 \pm 0.1$ | $0.01$ |

Table 1: Comparing with MCMC samplers for the constraint of including "amazing" in the sequence. We show mean $\pm$ one standard deviation over 10 runs. All samplers are run at an acceptance rate of $10^{-3}$. *TVD is estimated on $10^6$ independent samples, standard deviations are below 0.01. App. G contains additional results from these experiments.

distinct bigrams (Dist-2; Li et al., 2016). However, we note that these metrics can easily be cheated. An example of this is if we would manually construct a sampler that would repeat a small set of reasonably diverse, high probability (under GPT-2) sequences that meet the target constraint. The resulting sampling distribution would not be close to the target EBM, but would score well on such proxy metrics. For QRS we do report the TVD to the target distribution, which cannot be gamed in this way.[12] Notably, for MCMC we can also provide *lower bounds* on the TVD thanks to the data-processing inequality (DPI). The DPI tells us, informally, that the $f$-divergence of one distribution to another can only decrease by applying the same "projection" to the two distributions. A precise formulation of this theorem is provided as Theorem 6.2 of Polyanskiy and Wu (2017), but, for our purposes, the following special case ("lumping property") proven as Lemma 4.1 of Csiszár and Shields (2004) will be sufficient:

**Theorem 3.1.** *Let $p$ and $q$ be distributions over a sample space $X$. Let $\{A_1, \ldots, A_k\}$ be a finite partition of $X$. Then the distributions $p'(i) \doteq p(A_i)$ and $q'(i) \doteq q(A_i)$ over $\{1, \ldots, k\}$ satisfy $D_f(p', q') \leq D_f(p, q)$.*

While it is in general unfeasible to estimate the actual divergence of two distributions $p, q$ over a *large* space $X$ based only on a moderate number of *samples* (as opposed to *scores*) from $p$ and $q$ (Canonne, 2020), by projecting these samples onto a much smaller number $k$ of "bins", it is possible to obtain precise estimates of the divergence of the "histogram" $p'$ from $q'$, and therefore to *lower bound* the divergence of $p$ from $q$.

We use this result to obtain lower bounds on the TVD to the target distribution of RWMH-R and IMH-R, for which we have i.i.d. samples, whereas this is not the case for RWMH and IMH. After manually inspecting a single set of $10^4$ samples out of ten independently sampled sets to look for salient defects, we chose a binning function that classifies a sample in two bins according to whether it contains a newline character or not. Then, we used it to bin the samples of the remaining nine sets and compute their mean TVD to the binned target distribution to obtain lower bounds on the TVD.

**Results** Our results are shown in Table 1 (with some samples shown in App. G). All samplers satisfy the constraint of only generating sequences containing the term "amazing". The EBM $p$, however, is defined to be, among all those distributions that satisfy the constraint, the one closest to the original language model $a$ (GPT-2) under $D_{\mathrm{KL}}(p, a)$. Constraint satisfaction alone thus does not tell us how well the samplers approximate the target EBM. For MCMC samplers we have to rely on proxy metrics. RWMH-R seems to excel in terms of perplexity while also obtaining competitive diversity metrics (Self-BLEU-5, % Uniq, and Dist-2). However, we can identify a large TVD between its sampling distribution and the target one, showcasing the failure of proxy metrics to reliably measure the approximation accuracy of our samplers, and

---

[12]This may raise a question: Could one design *any* proxy metrics that would *reliably* assess closeness of an MCMC sampler to the target $p$, given a limited number of MCMC samples? When the space $X$ is very large, as is the case here, probably not. For an intuitive example, say $p$ is the uniform distribution over $X$ and $\omega$ is the uniform distribution over an unknown subset $Y$ of $X$, of size half that of $X$. The number of samples to distinguish these distributions clearly grows with the size of $X$ if *only samples* from $\omega$ are available, but if the value of the *probability* $\omega(x)$ is also available then a *single sample* $x$ is sufficient. More formally, one might consider algorithms that compare two distributions $p$ and $\omega$ on a set $X$ of size $n$, which should 'accept' with probability 2/3 if $\mathrm{TVD}(p, \omega) \leq \varepsilon_1$ and 'reject' with probability 2/3 if $\mathrm{TVD}(p, \omega) \geq \varepsilon_2$. Canonne (2020, Theorem 12.9, p. 58) shows that such algorithms require $O(1/(\varepsilon_1 - \varepsilon_2)^2)$ samples from $\omega$ if the probabilities $\omega(x)$ are known; whereas Canonne et al. (2022, Theorem 1.1) show that $\Omega\left(\frac{\sqrt{n}}{\varepsilon_2^2} + \frac{n}{\log n} \max\left\{\frac{\varepsilon_1}{\varepsilon_2^2}, \frac{\varepsilon_1^2}{\varepsilon_2^4}\right\}\right)$ samples are needed if these probabilities are unavailable.

demonstrating the benefits of having explicit estimates of divergence measures. After qualitatively inspecting samples of both local MCMC methods (see Tables 6 and 7 in App. G), we found many samples to contain repetitions, e.g. of punctuation marks. Also, RWMH seems to suffer from low diversity across samples (as evidenced by the high Self-BLEU-5 and low % Uniq scores), containing many repeated samples. On the other hand, IMH, IMH-R and QRS do not suffer from this lack of diversity. The variants of IMH and QRS seem to perform on par in terms of sample diversity, and QRS attains slightly lower perplexity than IMH-R.

An estimate of the TVD around 0.01 shows that QRS closely approximates the target EBM and shows considerably more diversity within and across samples. It might be that the IMH and IMH-R samplers would achieve similar results, but we cannot know as no divergence estimates are available. We only know that the binning strategy failed to detect any large divergences for IMH-R. One case in which we can estimate $f$-divergences for IMH-R and RWMH-R variants is the Poisson example that we discussed in Section 3.1. We present a comparison between QRS and those reset variants of MH for this Poisson example in Appendix C.2, finding that QRS consistently results in lower divergences for all acceptance rates (for $\mathrm{AR}_\beta < 1$).

## 4 Related Work

The vast majority of approaches to approximate sampling from complex probability distributions have been based on MCMC. However, a few approaches have taken rejection sampling as their starting point. Like QRS, the method of rejection sampling chains (Tierney, 1992; Chib and Greenberg, 1995, Sec. 6.1) does not require a global upper bound. This is a hybrid method that uses rejection sampling in a region satisfying a partial upper bound but combines it with IMH outside of that region to produce a Markov chain that converges to the correct stationary distribution. Caffo et al. (2002) propose empirical supremum rejection sampling, an algorithm that adaptively increases the $\beta$ upper bound based on the maximum observed so far, with a focus on convergence in the limit rather than approximation quality.

Closer to our work, some researchers have observed before us that a partial bound $\beta$ leads to the probability distribution presented in Eq. 3. Rejection control (Liu et al., 1998; Liu, 2004, pp. 44-45), in the context of particle filters, exploits this observation to accelerate the computation of an unbiased IS estimate of the expectation $\mathbb{E}_{x \sim p} f(x)$ in situations where computing $f(x)$ is expensive and must be done repeatedly. While the focus of that work was not to produce divergence diagnostics, interestingly, on close examination, we find that one of their proofs (Liu, 2004, App. A.1, Eq. (A.4)) formulated a $\chi^2$-divergence between a target distribution and a proxy distribution similar to $p_\beta$. Variational rejection sampling (Grover et al., 2018) uses a relaxation of Eq. 3 to better approximate the variational posterior in a variational inference setting. They aim to construct a differentiable sampler that can tighten the evidence lower bound, if additional computing power is available. The focus of our work is more general: we propose a generic sampler for which we can quantify a trade-off between sampling efficiency and approximation quality and we study the properties of this sampler.

While this paper is concerned with generating samples from *discrete* EBMs, much research so far has been more concerned with *continuous* EBMs, in particular for applications in vision. Continuous EBMs have the advantage over discrete ones that it is possible to differentiate the EBM $P(x)$ with respect to $x$, and not only the approximating model $\pi_\theta$ relative to $\theta$. This opens a range of optimized training techniques (see the survey by Song and Kingma 2021), including Langevin and Hamiltonian dynamics (Parisi, 1981; Duane et al., 1987), where the local Markov chain moves are informed by $\nabla_x \log P(x)$. While such techniques are not available for discrete EBMs, some recent efforts are trying to bridge the gap. For instance, *continuous relaxation* techniques (Han et al., 2020; Nishimura et al., 2020) relax the original discrete space into a continuous space, perform the sampling in this space, then map the samples back to the discrete space; while Grathwohl et al. (2021); Zhang et al. (2022); Rhodes and Gutmann (2022) sample directly in the discrete space, but inform the local moves of the chain through gradients computed in a larger continuous space. To the best of our knowledge, while Zhang et al. (2022) do experiment with "text infilling", the ability to replace some blanks in a given sentence by actual words, none of the above work has so far directly addressed text generation applications of the kind we have been considering here, in which the sample space is not only discrete, but composed of structured objects, namely word sequences of varying length, raising specific challenges.

Monte Carlo sampling techniques are popular for various NLP applications, in particular language modeling. For example, Miao et al. (2020) propose a sampler that mitigates poor estimation of probabilities due to overfitting. Deng et al. (2020) train globally normalized language models to combat negative effects of local normalization, and use a form of sampling importance resampling (Rubin, 1987) to sample from the resulting EBM using an autoregressive proposal language model. Goyal et al. (2021) develop a Metropolis-Hastings algorithm to sample from masked language models. For controlled text generation Miao et al. (2019) propose a random-walk Metropolis-Hastings algorithm for sampling from an EBM that encodes sequence-level preferences on natural text. Their proposal distribution consists of local string editing operations on randomly selected words or positions. Zhang et al. (2020) improve on this approach by making use of a tree-search algorithm to more efficiently explore the space of proposals, by allowing several edits in a single step of the MH algorithm. In contrast to QRS, none of these approaches attempt to directly estimate the divergence between the sampler and the target EBM, but rely on unreliable proxy metrics.

## 5 Discussion

QRS is a simple-yet-powerful technique: why has it not previously been promoted as a practical sampling method? One possible reason is the limited repertoire of *global* surrogates to complex distributions, apart from certain special cases in Euclidean domains (e.g. log-concave targets). This lack has strongly motivated the development of MCMC techniques which can exploit simple *local* proposals to compute transition probabilities between samples. This is now rapidly changing with advances in neural training, such as the impressive ability of pretrained language models to exploit simple prompts to orient their productions towards certain desired outcomes (but without formal guarantees); or with certain recent *generic* techniques, such as DPG, for fine-tuning autoregressive models towards arbitrary EBMs (but without the ability to totally reproduce them).

We believe that, given such global proposals, QRS can be a strong competitor to MCMC approaches. In particular, QRS has strong theoretical guarantees, not shared by these approaches: i) the ability to estimate, for any value of the $\beta$ parameter, the divergence of the target EBM from the QRS sampler $p_\beta$, for any member of the large class of $f$-divergences, including TVD and KL, ii) the ability to *tune* the sampler to attain a desired quality-efficiency trade-off, and the intuitive nature of this tuning process thanks to the monotonic relationship between parameter $\beta$ and the $f$-divergence (Theorem 2.2), iii) the existence of a simple, intuitive bound on the TVD between the QRS sampler and the target, provided by Theorem 2.1, and iv) the fact that QRS directly produces i.i.d. samples, rather than the correlated samples of a typical MCMC method.

Our experimental results show that QRS achieves strong results on the task of controlled text generation, where for instance, the sampler achieves excellent debiasing of the language model for acceptance rates in the range $10^{-1}$ to $10^{-3}$. We show the versatility of the approach by applying it to different sources of global proposals: proposals over EBMs obtained by generic DPG-style fine-tuning; proposals based on handcrafted prompts; and for paraphrase generation (App. B), proposals based on round-trip translation.

Finally, when comparing QRS to variants of Metropolis-Hastings, we find QRS outperforms the local variants (RWMH and RWMH-R) and performs on par with the global variants (IMH and IMH-R), according to the proxy metrics available for all samplers. Our results on RWMH and RWMH-R, however, show how proxy metrics can be deceiving and do not give us a full picture of the approximation accuracy of our samplers. Therefore, we stress the importance of well-founded divergence measures and, in this work, have proposed a sampler for which we can estimate these.

## Acknowledgments

We would like to thank the reviewers for their thorough feedback and suggestions for improvements. Also thanks to Jos Rozen for his help with experiments, and to Stéphane Clinchant who, in discussion with one of the authors, suggested using prompting for the purpose of producing global proposals oriented towards a certain generation objective.

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

# A    Additional Controlled Text Generation Results

We perform constrained text generation for a variety of distributional and pointwise constraints. In particular, we constrain the GPT-2 biographies model to contain (a) 50% female biographies about scientists, (b) 50% female biographies about sports, or (c) 50% female biographies without additional constraint. Also, we constrain GPT-2 small to exclusively generate sequences containing (d) the term "amazing", or (e) the term "Wikileaks". For each of these tasks, we obtained a fine-tuned model using DPG, which serves both as a baseline and as a proposal $q$ that we can sample from. In the case of pointwise constraints, we also consider a *naive filter* sampler $q_{proj}$ in which the proposal distribution is directly projected onto the constraint manifold by filtering out all samples that do not match the constraint. This sampler also assigns well-defined probabilities to the sequences that it samples, so we can compute estimates of the TVD and $D_{KL}$ for it.

For each task, we again obtain 1M samples from the corresponding proposal, which we use to evaluate the proposal $q$, the projected proposal $q_{proj}$ (only for the pointwise constraint), and QRS sampling ($p_\beta$) for a range of $\beta$ values reported in Table 2. For all of these, we compute estimates of a number of metrics including those of Sec. 3.2.2 (i.e. TVD($p, p_\beta$), $D_{KL}(p, p_\beta)$, AR, reverse KL divergence from the base language model $D_{KL}(\cdot, a)$, and the moments of the features that we wish to control).

Our results are shown in Fig. 5. As expected, the upper bound on the TVD between $p_\beta$ and $p$, and the KL divergence of $p$ from $p_\beta$ both converge monotonically to zero as the acceptance rate decreases. For the constraints and corresponding proposal distributions shown here, it seems that an acceptance rate of $10^{-3}$ is sufficient to match the target EBM nearly perfectly. The feature moments also converge as the acceptance rate decreases, although in some cases the QRS sampler matches the target EBM so closely that small inaccuracies in the $\lambda$ values obtained from the EBM estimation procedure (which is from Khalifa et al. 2021) become apparent. As for the divergence of the QRS distribution from the original language model $D_{KL}(p_\beta, a)$, there is no obvious trajectory that it should follow other than a tendency to converge to the lowest possible value $D_{KL}(p, a)$ when all constraints are satisfied (by definition, the EBM $p$ is the distribution $c$ which minimizes $D_{KL}(c, a)$ among all distributions satisfying the constraints). Indeed, our results show that this metric is a non-monotonic function of AR. We observe that the moments computed downstream on QRS closely match the IS predictions, giving us confidence in the accuracy of those estimates. Finally, in the case of our pointwise "amazing" and "Wikileaks" constraints, we find that the naive filter strategy ($q_{proj}$) corresponds to running the QRS sampler at a high acceptance rate.

| Experiment | $\beta_{\min}$ | $\beta_{\max}$ |
|---|---|---|
| 50% female and 100% scientists | $1.0 \cdot 10^{-12}$ | $9.3 \cdot 10^6$ |
| 50% female and 100% sports | $1.0 \cdot 10^{-12}$ | $2.9 \cdot 10^7$ |
| 50% female | $4.0 \cdot 10^{-7}$ | $4.0 \cdot 10^3$ |
| 100% "amazing" | $1.0 \cdot 10^{-12}$ | $5.3 \cdot 10^1$ |
| 100% "Wikileaks" | $1.0 \cdot 10^{-12}$ | 6.0 |
| simple | $1.0 \cdot 10^{-12}$ | 3.8 |
| multiple | $1.0 \cdot 10^{-12}$ | 2.3 |
| knowledge | $1.0 \cdot 10^{-12}$ | 24.0 |
| jeopardy | $1.0 \cdot 10^{-12}$ | 29.0 |
| news | $1.0 \cdot 10^{-12}$ | 3.0 |

Table 2: We report the range of $\beta$ values used to obtain the range of acceptance rates in Fig. 3, 4, and 5.

# B    Applications to Paraphrase Generation

Inspired by Miao et al. (2019), we perform proof-of-concept experiments on paraphrase generation by framing the task in terms of conditional EBMs. Specifically, given a sentence $y$ to paraphrase, we define our EBM in terms of a language model $a(x) = \text{GPT-2}(x)$, and a pointwise constraint $b(x)$ given by a binary classifier that classifies a pair $(x, y)$ as a paraphrase if the cosine similarity between their sentence embeddings is

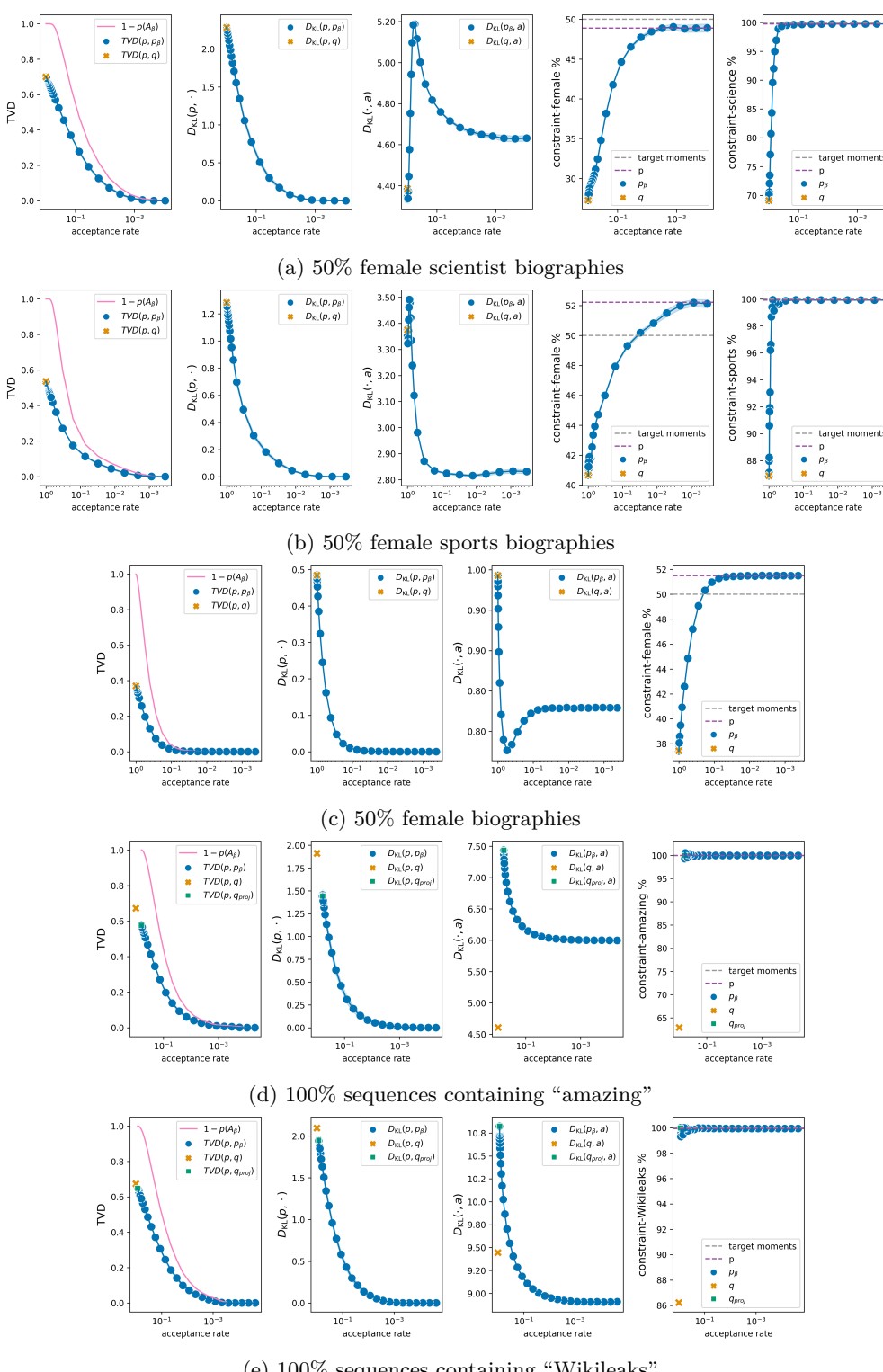

Figure 5: We show IS estimates of $\mathrm{TVD}(p, \cdot)$, an upper-bound on $\mathrm{TVD}(p, p_\beta)$, $D_{\mathrm{KL}}(p, \cdot)$, $D_{\mathrm{KL}}(\cdot, a)$ and feature moments as a function of acceptance rate. We show three distributional constraints on GPT-2 biographies and two pointwise constraints on GPT-2 small. As proposal distribution, we use a DPG model trained for each constraint separately. We show separate lines for the target moments and the moments realized by the EBMs, revealing slight inaccuracies in the EBM moments for some constraints.

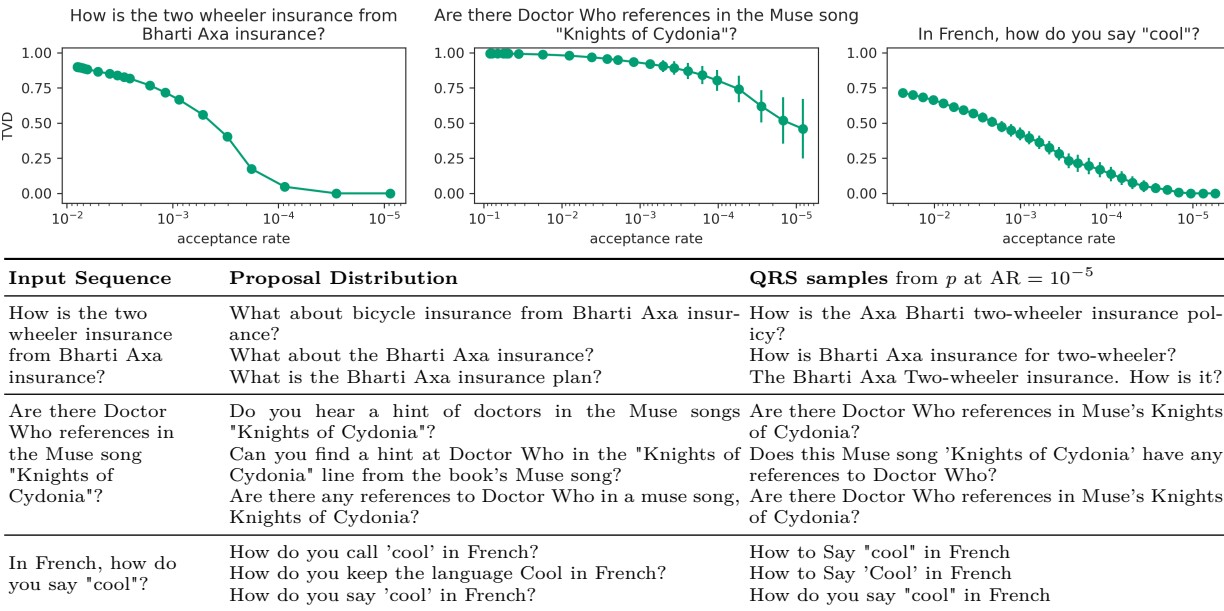

| Input Sequence | Proposal Distribution | QRS samples from $p$ at AR $= 10^{-5}$ |
|---|---|---|
| How is the two wheeler insurance from Bharti Axa insurance? | What about bicycle insurance from Bharti Axa insurance? 
 What about the Bharti Axa insurance? 
 What is the Bharti Axa insurance plan? | How is the Axa Bharti two-wheeler insurance policy? 
 How is Bharti Axa insurance for two-wheeler? 
 The Bharti Axa Two-wheeler insurance. How is it? |
| Are there Doctor Who references in the Muse song "Knights of Cydonia"? | Do you hear a hint of doctors in the Muse songs "Knights of Cydonia"? 
 Can you find a hint at Doctor Who in the "Knights of Cydonia" line from the book's Muse song? 
 Are there any references to Doctor Who in a muse song, Knights of Cydonia? | Are there Doctor Who references in Muse's Knights of Cydonia? 
 Does this Muse song 'Knights of Cydonia' have any references to Doctor Who? 
 Are there Doctor Who references in Muse's Knights of Cydonia? |
| In French, how do you say "cool"? | How do you call 'cool' in French? 
 How do you keep the language Cool in French? 
 How do you say 'cool' in French? | How to Say "cool" in French 
 How to Say 'Cool' in French 
 How do you say "cool" in French |

Figure 6: TVD($p, p_\beta$) running the QRS sampler at various acceptance rates to generate paraphrases of three sequences (**top**). We show some example paraphrases from both the proposal distribution $q(x)$ (round-trip NMT) as well as the QRS sampler $p_\beta$ at an acceptance rate of $10^{-5}$ (**bottom**).

above 0.95. We obtain high-quality sentence embeddings from sentence-BERT[13] (Reimers and Gurevych, 2019). As proposal distribution we do not use GPT-2, but rather illustrate how we can utilize off-the-shelf deep learning models as proposal distributions for QRS. In particular, we use a round-trip machine-translation model, which is a well-known tool in generating paraphrases (Bannard and Callison-Burch, 2005; Mallinson et al., 2017). Specifically, we use the English-to-German and German-to-English models from Ng et al. (2019). We first obtain a beam-searched (Graves, 2012) translation into German,[14] and then define the proposal distribution as the German-to-English model conditioned on the beam searched translation. We locally renormalize the model to do top-30 sampling (Fan et al., 2018).

We show IS estimates of TVD($p, p_\beta$) using 1M samples for three sequences in Fig. 6 along with samples from both the proposal distribution and QRS at AR $= 10^{-5}$. The quality of the proposal distribution varies with the input sequence, as can be seen from the slope of the curve and the low-efficiency starting points of some curves (non-paraphrases are always rejected and so they have a big influence on the acceptance rate). Still, QRS gives excellent approximations to the target EBM in two out of the three examples (in the "insurance" and "cool" examples, the TVD is nearly zero), although the TVD in the "Doctor Who" example remains above 0.4 even for AR $= 10^{-5}$). Looking at the examples, we find that the proposal distribution produces decent paraphrases, but they are not always semantically equivalent or grammatically correct. The QRS samples are mostly semantically equivalent, though they still contain some mistakes ("Axa Bharthi" vs "Bharti Axa") and seem to be insensitive to the question mark and to the casing of words ("Cool", "Two-wheeler insurance"). Interestingly, this experiment illustrates how the presented approach could be employed to *disentangle* the questions of how to model a problem (by defining the corresponding EBM) and how to efficiently sample from it (by improving the proposal distributions), making it possible to work on each of these questions separately.

---

[13]We use https://huggingface.co/sentence-transformers/all-mpnet-base-v2.
[14]We use a beam size of 5.

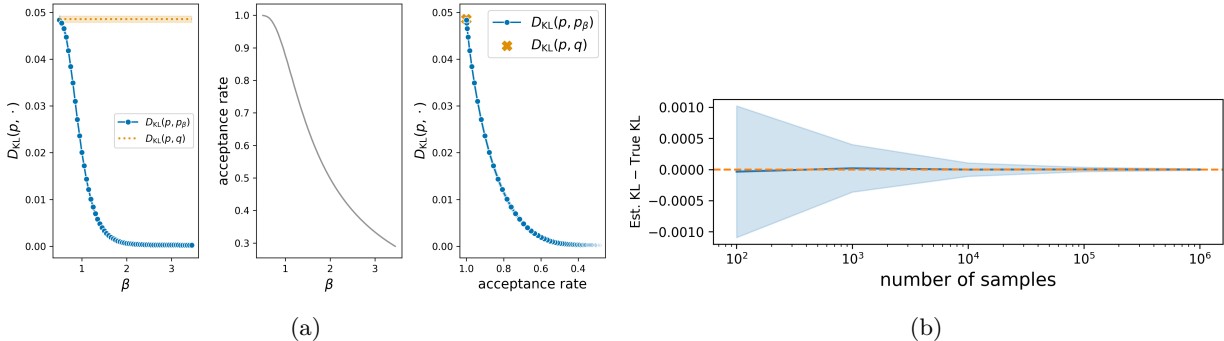

Figure 7: (a) Estimation of sampling quality as $D_{\mathrm{KL}}(p, p_\beta)$, efficiency (acceptance rate), and the trade-off between them for a QRS sampler when using a proposal $q = \mathrm{Poisson}(\lambda = 10)$ to approximate $p = \mathrm{Poisson}(\lambda = 11)$, computed in 10 independent experiments over $10^4$ samples. (b) Differences between estimated $D_{\mathrm{KL}}(p, p_\beta)$ and their true values for $\beta = 2$, computed 1000 times for each different number of samples. Shaded areas represent one standard deviation.

## C  Additional Results for Poisson Distributions

### C.1  KL-divergence results for QRS

Fig. 7 shows the convergence of estimates of the KL divergence when using QRS with Poisson distributions. As in the analogous plots for TVD (Fig. 2), we see that the divergence quickly converges to zero as $\beta$ increases, and that the standard deviation of the estimates is much larger than the bias.

### C.2  Comparison of QRS, IMH-R and RWMH-R

We now compare the performance of QRS with other approximate sampling methods, namely the reset versions of the independent Metropolis Hastings (IMH-R) and the random-walk Metropolis Hastings (RWMH-R) samplers. As above, we use the proposal $q = \mathrm{Poisson}(\lambda = 10)$ and target $p = \mathrm{Poisson}(\lambda = 11)$. To ensure accurate values for the KL divergence and TVD, we implement each method in closed form, as now explained.

**Calculating the probability mass functions of the samplers**  For QRS, we compute the probability mass function $p_\beta$ and acceptance rate using Eq. 3 and 4. For IMH-R, we take $k$ samples from the proposal $q$ according to the generative process:

$$X^{(0)} \sim q,$$
$$\textbf{for } t = 1, \ldots, k - 1$$
$$\quad Y^{(t)} \sim q$$
$$\quad X^{(t)} = \begin{cases} Y^{(t)} & \text{with probability } \min\left(1, \frac{p(Y^{(t)})/q(Y^{(t)})}{p(X^{(t-1)})/q(X^{(t-1)})}\right) \\ X^{(t-1)} & \text{otherwise.} \end{cases}$$

For nonnegative integer states $X^{(t-1)} = x_0$ and $X^{(t)} = x_1$, this corresponds to the transition matrix (or Markov kernel, noting that the state space is infinite)

$$T_{x_1, x_0} = q(x_1) \min\left(1, \frac{p(x_1)/q(x_1)}{p(x_0)/q(x_0)}\right), \qquad \forall x_1 \neq x_0$$
$$T_{x_0, x_0} = 1 - \sum_{x_1 \neq x_0} T_{x_1, x_0}.$$

We compute the divergence of the target from the distribution of $X^{(k-1)}$, which is $T^{k-1}q$; and for comparison with QRS, we compute the "acceptance rate" as $1/k$, i.e. the reciprocal of the number of times we sample

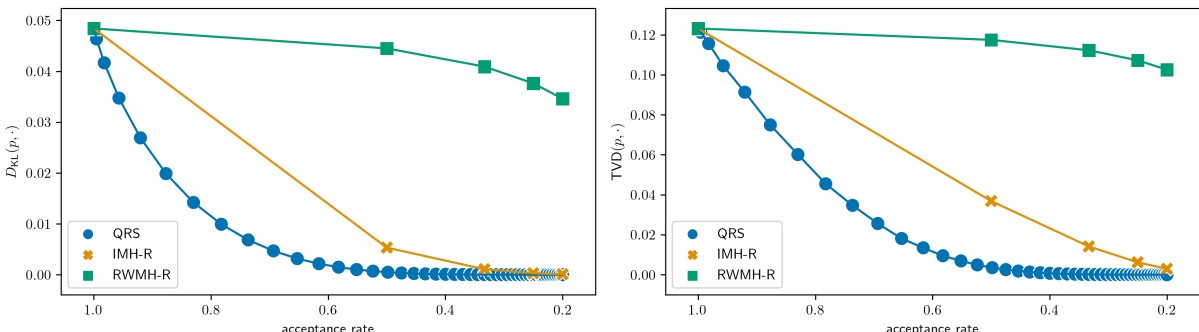

Figure 8: Quality-efficiency trade-off for QRS, IMH-R and RWMH-R samplers, when approximating the target distribution $p = \mathrm{Poisson}(\lambda = 11)$. We use distribution $q = \mathrm{Poisson}(\lambda = 10)$ as proposal for QRS and IMH-R, and as initial distribution for RWMH-R. Quality is measured as $\mathrm{TVD}(p, p_{\mathrm{sampler}})$ in the left plot, and as $D_{\mathrm{KL}}(p, p_{\mathrm{sampler}})$ in the right plot. Efficiency is measured as the acceptance rate for QRS; and as the reciprocal of the number of sampling iterations for IMH-R and RWMH-R, which we call the "acceptance rate" in this context.

from $q$. For RWMH-R, we consider the chain:

$$X^{(0)} \sim q,$$
$$\textbf{for } t = 1, \ldots, k-1$$
$$Y^{(t)} \sim \begin{cases} X^{(t-1)} - 1 & \text{with probability } \frac{1}{2} \\ X^{(t-1)} + 1 & \text{otherwise} \end{cases}$$
$$X^{(t)} = \begin{cases} Y^{(t)} & \text{with probability } \min\left(1, \frac{p(Y^{(t)})}{p(X^{(t-1)})}\right) \\ X^{(t-1)} & \text{otherwise,} \end{cases}$$

corresponding to the transition matrix

$$T_{x_1, x_0} = \begin{cases} \min\left(1, \frac{p(x_1)}{p(x_0)}\right) & \text{if } |x_1 - x_0| = 1 \\ 0 & \text{if } |x_1 - x_0| > 1 \\ 1 - \sum_{x_1 : |x_1 - x_0| = 1} T_{x_1, x_0} & \text{if } x_1 = x_0. \end{cases}$$

We compute the divergence of the target from the distribution of $X^{(k-1)}$, which is $T^{k-1} q$; and for comparison with QRS, we compute the "acceptance rate" as $1/k$, i.e. the reciprocal of the number of times we must evaluate the target distribution $p$ (or for $k = 1$, the number of times we sample from the proposal $q$). Clearly, the performance of RWMH-R might be improved by using a different random walk: the $\pm 1$ random walk is used here as it seems the simplest and most natural choice.

**Truncation errors** For all methods, to ensure practical computation, we truncate the distributions, working only with states $x < 50$. To assess the truncation error, we repeat the experiment, this time truncating with $x < 100$, and find that the largest relative error in KL, TVD or AR for any method is in the KL divergence of QRS for $\beta = 5$, which is

$$\left| \frac{D_{\mathrm{KL}}(p, p_\beta)_{x<100} - D_{\mathrm{KL}}(p, p_\beta)_{x<50}}{D_{\mathrm{KL}}(p, p_\beta)_{x<100}} \right| = 1.48 \cdots \times 10^{-10}.$$

**Results** Figure 8 presents the TVD and KL divergence as a function of AR, computed for parameter $\beta$ in the range $[0.1, 5]$ for QRS, and for $k = 1, 2, \ldots 5$ iterations of IMH-R and RWMH-R. For acceptance rate AR $= 1$, all samplers give the same divergence, as they all return samples directly from the proposal $q$.

For each lower acceptance rate[15], the TVD of QRS is systematically lower than that of IMH-R, and the TVD of IMH-R is systematically lower than that of RWMH-R; and the same holds for KL divergence. On investigating the ratios of the divergences, we find that these advantages increase as the acceptance rate decreases, and for AR $\approx 1/5$ we have

$$\frac{\text{TVD}(p, p_{\text{IMH-R},k=5})}{\text{TVD}(p, p_{\text{QRS},\beta=5})} = 2.24\cdots \times 10^3 \qquad \text{and} \qquad \frac{D_{\text{KL}}(p, p_{\text{IMH-R},k=5})}{D_{\text{KL}}(p, p_{\text{QRS},\beta=5})} = 3.60\cdots \times 10^2.$$

## D Properties of QRS: Proofs

### D.1 Proofs of Eq. 3 and Eq. 4

**Proof of Eq. 3**   By definition, $p_\beta(x)$ is the probability that the first[16] output from Algorithm 1 is equal to $x$. On the first step of the algorithm, the probability that a given $x$ is accepted is $q(x)r_x$, whereas the probability that the algorithm rejects its first sample is $\rho \doteq \sum_{x \in X} q(x)(1 - r_x) = 1 - \sum_{x \in X} q(x)r_x$. More generally, the probability for $x$ to be accepted on step $i$ of the algorithm, while no $x$ was accepted on previous steps is then $\rho^{i-1}q(x)r_x$. Overall, the probability $p_\beta(x)$ that the first sample accepted is $x$ is

$$\sum_{i=1}^{\infty} \rho^{i-1}q(x)r_x = q(x)r_x \sum_{i=1}^{\infty} \rho^{i-1} = \frac{1}{1-\rho}q(x)r_x$$

$$= \frac{1}{\sum_{x \in X} q(x)r_x}q(x)r_x = \frac{1}{Z_\beta}P_\beta(x).$$

**Proof of Eq. 4**   We have

$$\text{AR}_\beta = \mathbb{E}_{x \sim q}\min(1, P(x)/(\beta q(x)))$$

$$= \beta^{-1}\sum_{x \in X}\min(P(x), \beta q(x))$$

$$= \beta^{-1}\sum_{x \in X}P_\beta(x) = Z_\beta/\beta.$$

### D.2 Proof of Theorem 2.1

We will need a well-known property of TVD (e.g. see Chafaï 2010), namely that, for any distributions $p_1, p_2$ over $X$, we have

$$\text{TVD}(p_1, p_2) = \sum_{x \in X: p_1(x) \geq p_2(x)} (p_1(x) - p_2(x)). \tag{8}$$

The proof, illustrated in Fig. 9, then proceeds as follows:

Let $A_\beta \doteq \{x \in X : P(x) \leq \beta q(x)\}$ and $\bar{A}_\beta \doteq X \setminus A_\beta$. We have $P_\beta(x) \doteq \min(P(x), \beta q(x))$ and therefore $P_\beta(x) = P(x)$ for $x \in A_\beta$, and $P_\beta(x) < P(x)$ for $x \in \bar{A}_\beta$. Overall, $P_\beta$ is less than or equal to $P$, and thus $Z_\beta \leq Z$. For any $x$, we have $p_\beta(x) = P_\beta(x)/Z_\beta$ and $p(x) = P(x)/Z$, and hence for $x \in A_\beta$ we have $p(x) \leq p_\beta(x)$.

If we define $C_\beta \doteq \{x \in X : p(x) \leq p_\beta(x)\}$, and $\bar{C}_\beta \doteq X \setminus C_\beta$, we have $A_\beta \subseteq C_\beta$ and $\bar{C}_\beta \subseteq \bar{A}_\beta$. Using Eq. (8), we now see that

$$\text{TVD}(p, p_\beta) = \sum_{x \in X: p(x) \geq p_\beta(x)} (p(x) - p_\beta(x)) = \sum_{x \in \bar{C}_\beta} (p(x) - p_\beta(x)) \leq \sum_{x \in \bar{C}_\beta} p(x).$$

Finally we get

$$\text{TVD}(p, p_\beta) \leq p(\bar{C}_\beta) \leq p(\bar{A}_\beta) = 1 - p(A_\beta). \quad \square \tag{9}$$

---

[15]The acceptance rates of QRS for $\beta = 2, 3, \ldots$ are nearly equal to $1/2, 1/3, \ldots$ because $\text{AR} = Z_\beta/\beta$ by Eq. 4, and for this choice of $\beta$, $p$ and $q$ we have $Z_\beta \approx 1$.

[16]Or, for that matter, due to the obvious i.i.d. nature of the algorithm, for any fixed $k$, the $k$-th output.

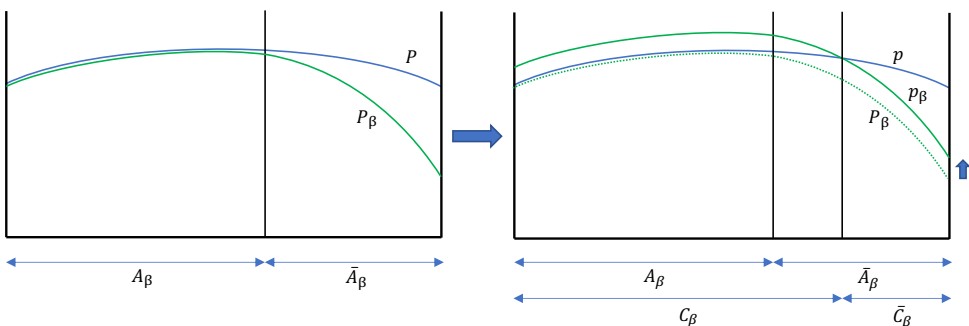

Figure 9: Visualization of Theorem 2.1. The left panel shows the unnormalized distributions $P$ and $P_\beta$, the right panel their normalized versions $p$ and $p_\beta$. In the right panel, the area under the curves $p$ and $p_\beta$ represent the total $p$-mass and $p_\beta$-mass of $X$ respectively, which are both 1. To simplify visual comparison, the figure assumes that $Z = 1$, in other words that $P = p$; then $Z_\beta \leq 1$ and $p_\beta$ is $P_\beta$ scaled by the factor $1/Z_\beta$. The TVD between $p$ and $p_\beta$ is equal to the area between the two curves above $C_\beta$, but also to the area between the two curves above $\bar{C}_\beta$. This last area is included inside the area below the $p$ curve above $\bar{A}_\beta$, which is the visual counterpart of Eq. (9).

**Proof of the corollary to the theorem** For any (normalized) distribution $p$ over a discrete space $X$, we have $\sum_{i=1}^{\infty} p(x_i) = 1$, hence for any $\epsilon > 0$, there exists a finite subset $X' \subseteq X$ such that $p(X') > 1 - \epsilon$, with $X' \subseteq \text{Supp}(p) \subseteq \text{Supp}(q)$. If we take $\beta \doteq \max_{x \in X'} \frac{P(x)}{q(x)}$, then $\beta$ is finite, $X' \subseteq A_\beta$, and therefore $p(A_\beta) \geq 1 - \epsilon$, which proves the result.

**A generalization to arbitrary $q$-supports** The corollary exploits the assumption — that we made throughout Sec. 2 — that the support of $p$, $\text{Supp}(p)$, is contained in the support of $q$, $\text{Supp}(q)$, in other terms, $p(x) > 0 \Rightarrow q(x) > 0$. If that were not the case then $\beta \doteq \max_{x \in X'} \frac{P(x)}{q(x)}$ could be infinite and $A_\beta$ would not be defined. However, it is interesting that we can actually generalize the result to the case where $\text{Supp}(p)$ may *not* be contained in $\text{Supp}(q)$. In that case, for any $\epsilon > 0$, there exists a finite subset $Y' \subseteq \text{Supp}(q)$ such that $p(Y') > p(\text{Supp}(q)) - \epsilon$. If we now take $\beta \doteq \max_{x \in Y'} \frac{P(x)}{q(x)}$, then $\beta$ is finite, $Y' \subseteq A_\beta$, and therefore $p(A_\beta) \geq p(\text{Supp}(q)) - \epsilon$. For any $\beta$, each element $x$ of $A_\beta$ is either in $\text{Supp}(q)$, or it satisfies $0 = \beta q(x) \geq P(x)$ so that $p(x) = 0$. It follows that $p(A_\beta) \leq p(\text{Supp}(q))$. Therefore

$$p(\text{Supp}(q)) \geq p(A_\beta) \geq p(\text{Supp}(q)) - \epsilon.$$

When the support of $p$ is included in $\text{Supp}(q)$, we have $p(\text{Supp}(q)) = 1$, and therefore we get the previous result back, but now we see that, in the general case

$$\lim_{\beta \to \infty} (1 - p(A_\beta)) = 1 - p(\text{Supp}(q)). \tag{10}$$

### D.3   Proof of Theorem 2.2 (Monotonicity)

**Notation** It will be convenient to use the notation $a \wedge b$ to denote $\min(a, b)$.

#### D.3.1   Preliminaries: majorization of convex functions and a squeezing lemma

We will exploit the following "majorization" result about convex functions.

**Theorem D.1** (Fuchs 1947). *Let $p_1, \ldots, p_n$ be real numbers, and let*

$$b > y_1 \geq \ldots \geq y_n > a, \qquad b > y_1' \geq \ldots \geq y_n' > a$$

*be nonincreasing sequences of real numbers, such that*

$$\sum_{i=1}^{n} p_i y_i' = \sum_{i=1}^{n} p_i y_i, \qquad \sum_{i=1}^{k} p_i y_i' \leq \sum_{i=1}^{k} p_i y_i, \tag{11}$$

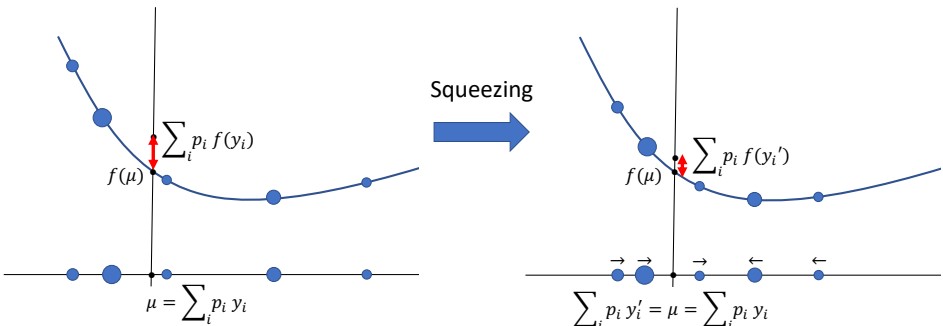

Figure 10: Squeezing lemma. We let $\mu = \sum_i p_i y_i$, and represent the points $y_i$ and $y'_i$ through dots with sizes proportional to $p_i$.

*for $k = 1, \ldots, n - 1$. Then for any convex function $f : (a, b) \to \mathbb{R}$, we have*

$$\sum_{i=1}^{k} p_i f(y'_i) \le \sum_{i=1}^{k} p_i f(y_i). \tag{12}$$

Our application of this theorem will be based on the following "squeezing" lemma (see Fig 10).

**Lemma D.2.** *Let $p_1, \ldots, p_n$ be nonnegative real numbers, with $\sum_{i=1}^{n} p_i = 1$, let $f$ be a convex function $f : (a, b) \to \mathbb{R}$, and let*

$$b > y_1 \ge \ldots \ge y_n > a, \qquad b > y'_1 \ge \ldots \ge y'_n > a$$

*be nonincreasing sequences of real numbers, such that*

$$\sum_{i=1}^{n} p_i y'_i = \sum_{i=1}^{n} p_i y_i.$$

*Suppose that there exists an integer $k$ such that $y'_i \le y_i$ if $i \le k$ and $y'_i \ge y_i$ otherwise. Then*

$$\sum_{i=1}^{n} p_i f(y'_i) \le \sum_{i=1}^{n} p_i f(y_i). \tag{13}$$

*Proof.* If $m \le k$ then $\sum_{i=1}^{m} p_i(y_i - y'_i) \ge 0$, as $y'_i \le y_i$ for $i \le k$. If $m > k$ then

$$\sum_{i=1}^{m} p_i(y_i - y'_i) = \sum_{i=m+1}^{n} p_i(y'_i - y_i) \ge 0,$$

as $\sum_{i=1}^{n} p_i(y_i - y'_i) = 0$ and $y'_i \ge y_i$ for $i > k$. This means that the conditions (11) are satisfied, and we can then conclude through (12). $\qquad \square$

### D.3.2 For the main proof, we can focus on the case where $P$ is normalized, the general case follows

Suppose that $p$ is the normalized version of $P$, with $p(x) = P(x)/Z$. We have by definition $p_\beta(x) = (P(x) \wedge (\beta q(x)))/Z_\beta$, in other words $p_\beta(x) \propto P(x) \wedge (\beta q(x))$. Let us denote by $\tilde{p}_{\tilde{\beta}}$ the similar distribution,

but this time defined relative to $\tilde{P} \doteq p$, in other words $\tilde{p}_{\tilde{\beta}}(x) \propto \tilde{P}(x) \wedge (\tilde{\beta}\, q(x)) = p(x) \wedge (\tilde{\beta}\, q(x))$, or equivalently, $\tilde{p}_{\tilde{\beta}}(x) \propto (Z\, p(x)) \wedge (Z\tilde{\beta}\, q(x)) = P(x) \wedge (Z\tilde{\beta}\, q(x))$, thus if $\beta = Z\tilde{\beta}$ then it follows that $p_\beta = \tilde{p}_{\tilde{\beta}}$. Therefore, in order to prove that $0 < \beta < \beta' < \infty$ implies $D_f(p, p_{\beta'}) \leq D_f(p, p_\beta)$, it is enough to prove that $0 < \tilde{\beta} < \tilde{\beta}' < \infty$ implies $D_f(p, \tilde{p}_{\tilde{\beta}'}) \leq D_f(p, \tilde{p}_{\tilde{\beta}})$.

### D.3.3   Some basic properties of $p_\beta$

Let $0 < \beta < \infty$. We have

$$P_\beta(x) = p(x) \wedge (\beta q(x)), \tag{14}$$

so that

$$Z_\beta = \sum_x P_\beta(x) = \sum_x p(x) \wedge (\beta q(x)) = \sum_x p(x)\, (1 \wedge (\beta\, q(x)/p(x))), \tag{15}$$

$$p_\beta(x) = \frac{P_\beta(x)}{Z_\beta} = \frac{p(x) \wedge (\beta\, q(x))}{Z_\beta} = p(x)\frac{1 \wedge (\beta\, q(x)/p(x))}{Z_\beta}. \tag{16}$$

The following observations will be key in the proof. For $0 < \beta \leq \beta' < \infty$, we have

$$\frac{1}{Z_\beta} \geq \frac{1}{Z_{\beta'}}, \qquad \frac{\beta}{Z_\beta} \leq \frac{\beta'}{Z_{\beta'}}, \tag{17}$$

which are simple consequences of the following facts:

$$Z_\beta = \sum_x p(x) \wedge (\beta q(x)) \leq \sum_x p(x) \wedge (\beta' q(x)) = Z_{\beta'},$$

$$\frac{Z_\beta}{\beta} = \sum_x \frac{p(x)}{\beta} \wedge q(x) \geq \sum_x \frac{p(x)}{\beta'} \wedge q(x) = \frac{Z_{\beta'}}{\beta'}.$$

### D.3.4   Using the form $D_f(p_\beta, p)$ instead of the form $D_f(p, p_\beta)$

As already remarked in footnote 5, $f$-divergences have the property that $D_f(p_1, p_2) = D_{\tilde{f}}(p_2, p_1)$, where $\tilde{f}(t) \doteq tf(1/t)$ is the "perspective" transform of $f$, which is convex when $f$ is convex (Polyanskiy, 2019, Remark 7.2). [17]

For the monotonicity proof, it will be more convenient to use the form $D_f(p_\beta, p)$ than the form $D_f(p, p_\beta)$. We will then prove that for any convex function $f$ (with the usual conditions on $f$-divergences), if $0 < \beta < \beta' < \infty$ then $D_f(p_{\beta'}, p) \leq D_f(p_\beta, p)$, which will imply that for any convex function $f$ (with the usual conditions on $f$-divergences), if $0 < \beta < \beta' < \infty$ then $D_f(p, p_{\beta'}) \leq D_f(p, p_\beta)$, the original formulation of Theorem 2.2.

With $p$ fixed, we will use the abbreviation $D_\beta \doteq D_f(p_\beta, p)$, so that

$$D_\beta = D_f(p_\beta, p) = \sum_x p(x)\, f\left(\frac{p_\beta(x)}{p(x)}\right) = \sum_x p(x)\, f\left(\frac{1 \wedge (\beta\, q(x)/p(x))}{Z_\beta}\right). \tag{18}$$

### D.3.5   Focusing on the finite case $X = \{x_1, \ldots, x_m\}$ first

Our proof will now proceed by assuming that $X = \{x_1, \ldots, x_m\}$ is finite, but we will show at the end (Sec. D.3.8) that the proof generalizes to any countable space $X$.

---

[17]For discrete distributions $p_1, p_2$ in particular, this is always true, irrespective of whether $p_1 \ll p_2$ or $p_2 \ll p_1$ — with $\ll$ denoting absolute continuity, that is $p_1 \ll p_2$ meaning that $\mathrm{Supp}(p_1) \subseteq \mathrm{Supp}(p_2)$ and similarly for the reverse case — but this requires a small technical extension of the notion of $f$-divergence, see Polyanskiy (2019), Remark 7.1. In the case where $p_1 = p_\beta, p_2 = p$, we do not need this extension because we have both $p_\beta \ll p$ and $p \ll p_\beta$, an easy consequence of the fact that $p \ll q$ by our hypotheses and of $p_\beta(x) = \frac{P_\beta(x)}{Z_\beta} = \frac{p(x) \wedge (\beta\, q(x))}{Z_\beta}$.

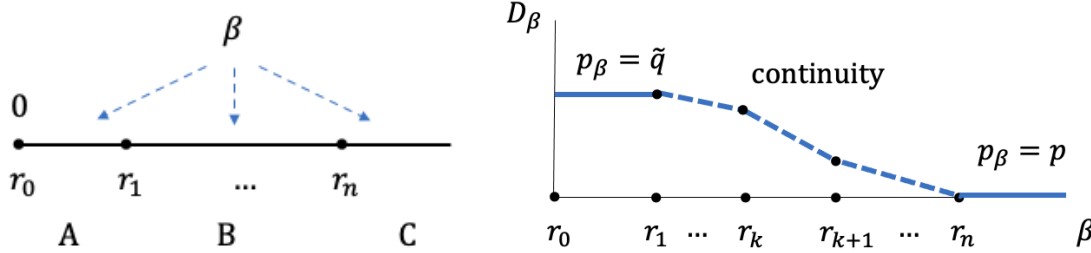

Figure 11: Left: three regions for $\beta$. Right: continuity of $D_\beta$.

### D.3.6 A reparametrization in terms of importance ratios

Let

$$r(x) \doteq p(x)/q(x),$$
$$R \doteq \{0\} \cup r(X) = \{r_0, r_1, \ldots, r_n\}$$

where $r_0 = 0$, and for $i > 0$, $r_i$ is the image by $r$ of some $x \in X$ (where $X = \{x_1, \ldots, x_m\}$ is now finite, by assumption; note that 0 may or not appear in $r(X)$ and that $m$ can be larger than $n+1$). We order the $r_i$ such that $r_i < r_{i+1}$:

$$\begin{array}{c} 0 \\ \bullet \quad \bullet \quad \cdots \quad \bullet \\ r_0 \quad r_1 \quad \cdots \quad r_n \end{array}$$

We can re-express Eq. 14–16 and 18 in terms of $r_i$:

$$p(r_i) \doteq \sum_{x \in X : r(x) = r_i} p(x), \tag{19}$$

$$Z_\beta = \sum_i p(r_i) \left(1 \wedge (\beta/r_i)\right), \tag{20}$$

$$p_\beta(r_i) \doteq p(r_i) \frac{1 \wedge (\beta/r_i)}{Z_\beta}, \tag{21}$$

$$D_\beta = \sum_i p(r_i) \, f\left(\frac{p_\beta(r_i)}{p(r_i)}\right) = \sum_i p(r_i) \, f\left(\frac{1 \wedge (\beta/r_i)}{Z_\beta}\right). \tag{22}$$

**Note 1** We abuse notation: $x \in X$ and $r \in \mathbb{R}_{\geq 0}$ are in different spaces, so we could really write $\breve{p}(r_i)$ instead of $p(r_i)$ to distinguish the two spaces.

**Note 2** We let $p(r_0) \doteq 0$ in all cases.

### D.3.7 Focusing on sub-intervals is enough

As the right-hand side of Eq. 20 is a continuous and positive function of $\beta \in (0, \infty)$, the right-hand side of Eq. 22 is also continuous, and therefore $D_\beta$ is a continuous function of $\beta \in (0, \infty)$. The index $i$ runs over a finite set, hence to prove that $D_\beta$ is non-increasing in $\beta$ it is enough to prove that it is non-increasing on each interval $(r_k, r_{k+1})$.

It will be useful to distinguish three regions for $\beta$ (see left panel of Fig. 11), namely region $A$, with $\beta$ smaller than the smallest non-null importance ratio $r_1$; region $B$ with $\beta$ between $r_1$ and $r_n$; and region $C$ with $\beta$ larger than the largest importance ratio $r_n$.

We start by noting that $p_\beta$ is constant on $A$ and on $C$:

Figure 12: Application of squeezing lemma to $r_k < b < b' < r_{k+1}$.

- For $\beta \in A$ we have

$$p(x) \wedge (\beta q(x)) = \begin{cases} 0 & p(x) = 0 \\ \beta q(x) & p(x) > 0 \end{cases}$$

and it follows that

$$p_\beta(x) = \begin{cases} 0 & p(x) = 0 \\ \frac{q(x)}{q(\mathrm{Supp}(p))} & p(x) > 0, \end{cases}$$

which is the "normalized" version of $q$ relative to $\mathrm{Supp}(p)$.

- For $\beta \in C$ we have $p_\beta(x) \propto p(x) \wedge (\beta q(x)) = p(x)$. Hence $p_\beta = p$ (standard RS).

It follows that $D_\beta$ is constant for $\beta \in A$, and again constant (actually null) for $\beta \in C$. Thus, in order to prove our monotonicity result, it is sufficient to prove that $D_\beta$ decreases for $\beta$ increasing inside each interval $(r_k, r_{k+1})$ for $k = 1, 2, \ldots, n-1$ (see right panel of Fig. 11).

Let $1 \le k < n$ and consider $r_k < b < b' < r_{k+1}$, we want to show that $D_{b'} = D_f(p_{b'}, p) \le D_b = D_f(p_b, p)$. For $i = 1, 2, \ldots, n$, let

$$y_i \doteq \frac{1 \wedge \frac{b}{r_i}}{Z_b}, \qquad y_i' \doteq \frac{1 \wedge \frac{b'}{r_i}}{Z_{b'}}. \tag{23}$$

From (20) and (22), we have

$$\sum_i p(r_i)\, y_i = 1, \qquad \sum_i p(r_i)\, y_i' = 1, \tag{24}$$

$$D_b = \sum_i p(r_i) f(y_i), \qquad D_{b'} = \sum_i p(r_i) f(y_i'). \tag{25}$$

We further note that

$$y_i = \begin{cases} \frac{1}{Z_b} & i \le k \\ \frac{b/r_i}{Z_b} & i > k \end{cases} \qquad \text{and} \qquad y_i' = \begin{cases} \frac{1}{Z_{b'}} & i \le k \\ \frac{b'/r_i}{Z_{b'}} & i > k, \end{cases} \tag{26}$$

and we saw earlier in (17) that

$$\frac{1}{Z_{b'}} \le \frac{1}{Z_b} \qquad \text{and} \qquad \frac{b'}{Z_{b'}} \ge \frac{b}{Z_b}. \tag{27}$$

It follows that

$$y_i' \le y_i \text{ for } i \le k \qquad \text{and} \qquad y_i \le y_i' \text{ for } i > k. \tag{28}$$

**We can now apply our squeezing lemma,** see Fig. 12: we apply Lemma D.2 to the $y_i, y_i'$ and $k$ of the previous paragraph, and let $p_i = p(r_i)$. By Eq. (24), (25) and (28), the hypotheses of the lemma are satisfied, and we conclude that $\sum_{i=1}^n p_i f(y_i') \leq \sum_{i=1}^n p_i f(y_i)$, so that

$$D_f(p_{\beta'}, p) \leq D_f(p_\beta, p). \tag{29}$$

### D.3.8 Concluding the proof: from the finite case to the (countably) infinite case

**(A)** Having proved the monotonicity result (29) for finite sample spaces $X$, we now wish to generalize this result to an arbitrary countable space $X$. We will use the following theorem, adapted from Gilardoni (2009, Proposition 1) to the case of probabilities over discrete spaces. For a partition $\mathcal{E} = \{E_1, \ldots, E_I\}$ of $X$ into $I$ subsets, and a distribution $p$ over $X$, let $p_{\mathcal{E}}$ denote the projection of $p$ onto the finite set $\{1, \ldots, I\}$ given by $p_{\mathcal{E}}(i) \doteq p(E_i)$.

**Theorem D.3** (Gilardoni 2009). *Let $r$ and $s$ be probability distributions over $X$. Let $D_f$ denote the $f$-divergence for some convex function $f$ with $f(1) = 0$. Then*

$$D_f(r, s) = \sup_{\mathcal{E}} D_f(r_{\mathcal{E}}, s_{\mathcal{E}}),$$

*where the* $\sup$ *is taken over all finite partitions $\mathcal{E}$ of $X$.*

**(B)** Let $p, q$ and $p_\beta$ be distributions over a countable space $X$, and let $a$ and $b$ be values of the QRS parameter $\beta$ such that $0 < a < b < \infty$. We prove that $D_f(p, p_b) \leq D_f(p, p_a)$ by contradiction. Suppose that $D_f(p, p_b) > D_f(p, p_a)$. Then, by Theorem D.3, there exists a finite partition $\mathcal{E} = \{E_1, \ldots, E_I\}$ of $X$ such that

$$D_f(p_{\mathcal{E}}, (p_b)_{\mathcal{E}}) > D_f(p, p_a). \tag{30}$$

**(C)** For each $i \in \{1, \ldots, I\}$, let us split $E_i$ into three subsets

$$E_i^{(1)} \doteq \{x \in E_i : p(x) \leq a\,q(x) \leq b\,q(x)\},$$
$$E_i^{(2)} \doteq \{x \in E_i : a\,q(x) < p(x) \leq b\,q(x)\},$$
$$E_i^{(3)} \doteq \{x \in E_i : a\,q(x) \leq b\,q(x) < p(x)\},$$

and consider the refinement of partition $\mathcal{E}$ given by

$$\mathcal{F} \doteq \{F_1, \ldots, F_J\} \doteq \{E_i^{(j)} : 1 \leq i \leq I, \ 1 \leq j \leq 3, \ E_i^{(j)} \neq \emptyset\},$$

so that $J \leq 3I$. Let $\beta \in \{a, b\}$. It is easy to see that $\mathcal{F}$ has the following *homogeneity* property: one can split the indices $Ind = \{1, \ldots, J\}$ into two disjoint subsets $Ind_\beta$ and $\overline{Ind_\beta}$, with

$$Ind_\beta \doteq \{j \in Ind : p(x) \leq \beta q(x)\ \forall x \in F_j\} \qquad \text{and} \qquad \overline{Ind_\beta} \doteq \{j \in Ind : p(x) > \beta q(x)\ \forall x \in F_j\}.$$

(Sets $Ind_\beta$ and $\overline{Ind_\beta}$ do not necessarily form a partition of $Ind$ as one of these sets may be empty.) We can now consider the projections $p_{\mathcal{F}}$ and $q_{\mathcal{F}}$ of $p$ and $q$ onto $Ind$, and the associated QRS distribution $(p_{\mathcal{F}})_\beta$, which is given by

$$(p_{\mathcal{F}})_\beta(j) \doteq \frac{p_{\mathcal{F}}(j) \wedge (\beta q_{\mathcal{F}}(j))}{\widetilde{Z_\beta}} = \frac{p(F_j) \wedge (\beta q(F_j))}{\widetilde{Z_\beta}} = \begin{cases} \frac{p(F_j)}{\widetilde{Z_\beta}} & j \in Ind_\beta \\ \frac{\beta q(F_j)}{\widetilde{Z_\beta}} & j \in \overline{Ind_\beta}, \end{cases}$$

where

$$\widetilde{Z_\beta} = \sum_{j \in Ind} p(F_j) \wedge (\beta\,q(F_j)) = \sum_{j \in Ind_\beta} p(F_j) + \sum_{j \in \overline{Ind_\beta}} \beta q(F_j).$$

But we also have

$$(p_\beta)_{\mathcal{F}}(j) = p_\beta(F_j) = \frac{\sum_{x \in F_j} p(x) \wedge (\beta q(x))}{Z_\beta} = \begin{cases} \frac{p(F_j)}{Z_\beta} & j \in Ind_\beta \\ \frac{\beta q(F_j)}{Z_\beta} & j \in \overline{Ind_\beta}, \end{cases}$$

where

$$Z_\beta = \sum_{x \in X} p(x) \wedge (\beta \, q(x)) = \sum_{j \in Ind_\beta} p(F_j) + \sum_{j \in \overline{Ind_\beta}} \beta \, q(F_j).$$

Comparing these two sets of equations, we see that $\widetilde{Z_\beta} = Z_\beta$ and that

$$(p_\beta)_{\mathcal{F}} = (p_{\mathcal{F}})_\beta.$$

**(D)** Let us now go back to Eq. 30. The distributions $p_{\mathcal{E}}$ and $(p_b)_{\mathcal{E}}$ can be seen as projections of the distributions $p_{\mathcal{F}}$ and $(p_b)_{\mathcal{F}}$ from the set $\{1, \dots, J\}$ to the set $\{1, \dots, I\}$, and therefore the DPI (Theorem 3.1) applies, implying that $D_f(p_{\mathcal{E}}, (p_b)_{\mathcal{E}}) \leq D_f(p_{\mathcal{F}}, (p_b)_{\mathcal{F}})$, and therefore:

$$D_f(p_{\mathcal{F}}, (p_b)_{\mathcal{F}}) > D_f(p, p_a).$$

A second application of the DPI theorem, this time projecting $p$ and $p_a$ from $X$ to the set $\{1, \dots, J\}$, implies that $D_f(p_{\mathcal{F}}, (p_a)_{\mathcal{F}}) \leq D_f(p, p_a)$, and therefore

$$D_f(p_{\mathcal{F}}, (p_b)_{\mathcal{F}}) > D_f(p_{\mathcal{F}}, (p_a)_{\mathcal{F}}).$$

**(E)** We saw earlier that $(p_b)_{\mathcal{F}} = (p_{\mathcal{F}})_b$ and $(p_a)_{\mathcal{F}} = (p_{\mathcal{F}})_a$, hence

$$D_f(p_{\mathcal{F}}, (p_{\mathcal{F}})_b) > D_f(p_{\mathcal{F}}, (p_{\mathcal{F}})_a),$$

which contradicts our monotonicity result for the finite set $\{1, \dots, J\}$, concluding the proof. $\qquad\square$

### D.4 Alternative proof of monotonicity for finite $X$, based on calculus

Sections D.3.1 to D.3.7 proved the monotonicity of the $f$-divergence $D_f(p_\beta, p)$ for distributions on a finite space $X$, based on a result about *majorization* (Theorem D.1). We now give an alternative proof, which avoids this potentially unfamiliar majorization result, and which is based on the easy-to-prove observation that the derivative of $D_f(p_\beta, p)$ with respect to $\beta$ is nonpositive. However, this derivative does not exist for all $\beta$, as neither the partition function $Z_\beta$ nor the function $f$ is necessarily differentiable. Thus to prove monotonicity, we must do some work to show that $D_f(p_\beta, p)$ is a Lipschitz continuous function for $\beta$ in an appropriate interval. We prove the following, which readily generalizes to infinite spaces $X$ using the arguments of Section D.3.8.

**Lemma D.4.** *Let $f : \mathbb{R}_{>0} \to \mathbb{R}$ be a convex function satisfying $f(1) = 0$. Let the set $X$ be finite, and let $0 < \beta < \beta' < \infty$. Then $D_f(p_{\beta'}, p) \leq D_f(p_\beta, p)$.*

*Proof.* The partition function $Z_b$ of the QRS distribution is

$$Z_b = \sum_i p(r_i)(1 \wedge (b/r_i))$$

as previously remarked in Eq. 20, where

$$R \doteq \{r_0, \dots, r_n\} \doteq \{0\} \cup \{p(x)/q(x) : x \in X\}.$$

If $b \notin R$ then this may be written in the form

$$Z_b = \sum_{\ell : r_\ell < b} p(r_\ell) + b \sum_{h : r_h > b} (p(r_h)/r_h),$$

so the quotient rule gives

$$\frac{d}{db}\frac{b}{Z_b} = \frac{Z_b - b\sum_{h:r_h>b}(p(r_h)/r_h)}{Z_b^2} = \frac{\sum_{\ell:r_\ell<b}p(r_\ell)}{Z_b^2}.$$

Similarly, as remarked in Eq. 22, the divergence is

$$D_f(p_b, p) = \sum_i p(r_i)f\left(\frac{1\wedge(b/r_i)}{Z_b}\right),$$

and if $b \notin R$ then this may be written in the form

$$D_f(p_b, p) = \sum_{\ell:r_\ell<b}p(r_\ell)f\left(\frac{1}{Z_b}\right) + \sum_{h:r_h>b}p(r_h)f\left(\frac{b}{r_h Z_b}\right).$$

Let $N$ be the set of values of $b \in [\beta, \beta']$ such that $f$ is nondifferentiable at one of the arguments $1/Z_b$ or $b/(r_h Z_b)$ in the above expression. For $b \notin R \cup N$, the chain rule gives

$$\frac{dD_f(p_b, p)}{db} = \sum_{\ell:r_\ell<b}p(r_\ell)f'\left(\frac{1}{Z_b}\right)\left(-\frac{\sum_{h:r_h>b}(p(r_h)/r_h)}{Z_b^2}\right) + \sum_{h:r_h>b}p(r_h)f'\left(\frac{b}{r_h Z_b}\right)\left(\frac{\sum_{\ell:r_\ell<b}p(r_\ell)}{r_h Z_b^2}\right)$$

$$= \sum_{\ell:r_\ell<b}\sum_{h:r_h>b}\frac{p(r_\ell)p(r_h)}{r_h Z_b^2}\left(f'\left(\frac{b}{r_h Z_b}\right) - f'\left(\frac{1}{Z_b}\right)\right) \le 0, \tag{31}$$

where the final inequality follows from the convexity of $f$.

Now we argue that $D_f(p_b, p)$ is a Lipschitz continuous function of $b \in [\beta, \beta']$.

- As the support of $q$ contains the support of $p$, we have $\max R < \infty$. Thus, the partition function $Z_b = \sum_i p(r_i)(1\wedge(b/r_i))$ is bounded by

$$\forall b \in [\beta, \beta'], \qquad 0 < Z_{\text{lo}} \doteq 1\wedge(\beta/\max R) \le Z_b \le 1.$$

- It follows that the arguments of function $f$ appearing in $D(p_b, p)$ are bounded by

$$\forall b \in [\beta, \beta'], \qquad 0 < Z_{\text{lo}} \le \frac{1\wedge(b/r_i)}{Z_b} \le \frac{1}{Z_{\text{lo}}}.$$

That is, all these arguments lie in an interval $[x_0, x_1]$ with $0 < x_0 < x_1 < \infty$.

- As $f$ is convex on $(0, \infty)$, the restriction of $f$ to the domain $[x_0, x_1]$ with $0 < x_0 < x_1 < \infty$ is Lipschitz.

- The following are Lipschitz: finite sums of Lipschitz functions; the minimum of two Lipschitz functions; the function $x \mapsto 1/x$ for $x > a$ where $a > 0$; the product of bounded Lipschitz functions; and the composition of two Lipschitz functions. It follows that $D(p_b, p)$ is Lipschitz for $b \in [\beta, \beta']$.

If a function is Lipschitz continuous on a closed bounded interval, then is differentiable almost everywhere on that interval, and the change in the function over that interval is equal to the integral of its derivative (see for instance Royden and Fitzpatrick 2010, Chapter 6, Proposition 7 and Theorem 10). Therefore,

$$D_f(p_{\beta'}, p) - D_f(p_\beta, p) = \int_\beta^{\beta'}\frac{dD_f(p_b, p)}{db}\,db \le 0,$$

where the inequality follows from Eq. 31. This completes the proof. $\qquad\square$

# E  Importance Sampling (IS) Estimates

Here we provide all relevant IS (Owen, 2013) estimates and their derivations. As mentioned in Sec. 2, we base all estimates on a sample $\{x_1, \ldots, x_N\}$ of i.i.d. draws from $q$; and if the sum $\sum_{x \in X} h(x)$ is well-defined for a function $h : X \to \mathbb{R}$, then we have

$$\sum_{x \in X} h(x) = \sum_{x \in X} q(x) \frac{h(x)}{q(x)} = \mathbb{E}_{x \sim q} \frac{h(x)}{q(x)} \simeq \frac{1}{N} \sum_{i=1}^{N} \frac{h(x_i)}{q(x_i)}.$$

In this way, we find

$$
\begin{aligned}
Z &= \sum_{x \in X} P(x) &&\simeq \frac{1}{N} \sum_{i=1}^{N} \frac{P(x_i)}{q(x_i)}, \\
Z_\beta &= \sum_{x \in X} P_\beta(x) &&\simeq \frac{1}{N} \sum_{i=1}^{N} \frac{P_\beta(x_i)}{q(x_i)}, \\
\mathrm{AR}_\beta &= \mathbb{E}_{x \sim q} \min\left(1, \frac{P(x)}{\beta q(x)}\right) &&\simeq \frac{1}{N} \sum_{i=1}^{N} \min\left(1, \frac{P(x_i)}{\beta q(x_i)}\right), \\
p(A_\beta) &= \sum_{x \in X} p(x) \mathbb{1}[x \in A_\beta] &&\simeq \frac{1}{N} \sum_{i=1}^{N} \frac{P(x_i)}{Z q(x_i)} \mathbb{1}[x_i \in A_\beta], \\
\mathbb{E}_{x \sim p_\beta} h(x) &= \sum_{x \in X} p_\beta(x) h(x) &&\simeq \frac{1}{N} \sum_{i=1}^{N} \frac{P_\beta(x_i)}{Z_\beta q(x_i)} h(x_i).
\end{aligned}
\tag{32}
$$

Similarly, we may estimate $f$-divergences with

$$D_f(p, p_\beta) = \mathbb{E}_{x \sim q} \frac{P_\beta(x)}{Z_\beta \, q(x)} f\left(\frac{Z_\beta \, P(x)}{Z \, P_\beta(x)}\right) \simeq \frac{1}{N} \sum_{i=1}^{N} \frac{P_\beta(x_i)}{Z_\beta \, q(x_i)} f\left(\frac{Z_\beta \, P(x_i)}{Z \, P_\beta(x_i)}\right),$$

which for TVD ($f(t) = |1 - t|/2$) and KL divergence ($f(t) = t \log t$) rearranges to

$$\mathrm{TVD}(p, p_\beta) \simeq \frac{1}{2N} \sum_{i=1}^{N} \left| \frac{P_\beta(x_i)}{Z_\beta \, q(x_i)} - \frac{P(x_i)}{Z q(x_i)} \right|, \tag{33}$$

$$D_{\mathrm{KL}}(p, p_\beta) \simeq \log \frac{Z_\beta}{Z} + \frac{1}{N} \sum_{i=1}^{N} \frac{P(x_i)}{Z q(x_i)} \log \frac{P(x_i)}{P_\beta(x_i)}. \tag{34}$$

# F  Estimating the Mapping Between $\beta$ and AR

Eq. 32 provides a way to estimate the AR given a value of $\beta$ and a set of samples from a proposal $q$. How can we go in the opposite direction and estimate $\beta$ for a target AR value? One way to do so, is to estimate the full mapping from AR to $\beta$, and interpolate it at the target AR. Algorithm 2 estimates this mapping efficiently, based on the observation that Eq. 32 can be rewritten as a sum of two terms:

$$\mathrm{AR}_\beta \simeq (a_i + b_i)/N \qquad \text{at } \beta = \beta_i \tag{35}$$

where

$$\beta_i \doteq P(x_i)/q(x_i), \qquad a_i \doteq \sum_{j=1}^{N} \mathbb{1}\left[\beta_j \le \beta_i\right] \beta_j/\beta_i, \qquad b_i \doteq \sum_{j=1}^{N} \mathbb{1}\left[\beta_j > \beta_i\right], \tag{36}$$

noting that both $a$ and $b$ can be computed efficiently given a sorted list of $\beta_i$ values. (In the case that $\beta_i = \beta_j$ for some $j \ne i$, the output will contain repeated values, which are easily filtered out, if necessary.)

| sampler | AR | %female | %science | PPL↓ | Self-BLEU-5↓ | %Uniq↑ | Dist-1↑ | Dist-2↑ | Dist-3↑ | TVD$(p,p_\beta)\downarrow^*$ | $D_{\text{KL}}(p,p_\beta)\downarrow^*$ |
|---|---|---|---|---|---|---|---|---|---|---|---|
| DPG | 1 | 27.3 ± 0.4 | 69.1 ± 0.4 | 34.4 ± 0.2 | 89.8 ± 0.1 | 100.0 ± 0 | 89.7 ± 0.1 | 95.8 ± 0.0 | 93.1 ± 0.0 | 0.7 ± 0.00099 | 2.28 ± 0.03 |
| IMH | $10^{-1}$ | 46.3 ± 4.3 | 99.8 ± 0.1 | - | 95.4 ± 0.5 | 55.7 ± 5.8 | 89.3 ± 0.7 | 96.0 ± 0.2 | 93.4 ± 0.2 | Unk. | Unk. |
| IMH-R | $10^{-1}$ | 40.9 ± 1.5 | 99.5 ± 0.2 | 29.2 ± 0.8 | 91.3 ± 0.1 | 100.0 ± 0 | 89.5 ± 0.2 | 95.9 ± 0.1 | 93.3 ± 0.0 | Unk. | Unk. |
| QRS | $10^{-1}$ | 44.2 ± 2.6 | 99.7 ± 0.1 | 29.9 ± 0.8 | 91.1 ± 0.2 | 100.0 ± 0 | 89.4 ± 0.2 | 96.0 ± 0.1 | 93.3 ± 0.0 | 0.37 ± 0.0049 | 0.77 ± 0.036 |
| RWMH-base | $10^{-1}$ | 29.9 ± 45.6 | 70.0 ± 45.8 | - | 100.0 ± 0.0 | 7.6 ± 15.1 | 89.3 ± 4.7 | 96.1 ± 1.0 | 93.0 ± 1.0 | Unk. | Unk. |
| RWMH-R-base | $10^{-1}$ | 28.0 ± 1.3 | 68.3 ± 1.5 | 34.9 ± 0.7 | 89.8 ± 0.2 | 100.0 ± 0 | 89.6 ± 0.2 | 95.8 ± 0.1 | 93.1 ± 0.1 | Unk. | Unk. |
| IMH | $10^{-3}$ | 49.8 ± 2.1 | 99.8 ± 0.1 | - | 91.2 ± 0.2 | 97.4 ± 1.0 | 89.1 ± 0.2 | 95.9 ± 0.1 | 93.4 ± 0.0 | Unk. | Unk. |
| IMH-R | $10^{-3}$ | 49.8 ± 1.6 | 99.8 ± 0.1 | 34.1 ± 0.8 | 90.9 ± 0.2 | 100.0 ± 0 | 89.1 ± 0.2 | 95.9 ± 0.1 | 93.4 ± 0.0 | Unk. | Unk. |
| QRS | $10^{-3}$ | 49.4 ± 1.4 | 99.8 ± 0.1 | 34.1 ± 0.9 | 90.9 ± 0.2 | 100.0 ± 0 | 89.1 ± 0.2 | 95.9 ± 0.1 | 93.4 ± 0.0 | 0.012 ± 0.0055 | 0.0084 ± 0.0048 |

Table 3: Further comparisons of samplers on the "female-science" EBM described in Sec. 3.2.2. We do not compute perplexity for IMH and RWMH without reset as it does not yield i.i.d. samples. As noted, TVD and KL for MCMC methods are unknown (i.e. we have no way of estimating them). Where available we show mean ± one standard deviation over 10 runs. *TVD and KL are estimated on independent sets of $10^6$ samples.

---

**Algorithm 2** Estimate AR $\to \beta$ mapping

---

1: **Require:** $P$, $q$, $N$
2: $S \leftarrow []$
3: **for** $i = 1, 2, \ldots, N$ **do**
4:      $x_i \sim q$
5:      $\beta_i \leftarrow P(x_i)/q(x_i)$
6:      $S[i] \leftarrow \beta_i$
7: **end for**
8: $S_s \leftarrow \texttt{SortAscending}(S)$             {Array of sorted $\beta_i$}
9: $a_{aux}[0] \leftarrow 0$
10: **for** $i = 1, 2, \ldots, N$ **do**
11:      $\beta_i \leftarrow S_s[i]$
12:      $a_{aux}[i] \leftarrow a_{aux}[i-1] + \beta_i$          $\{a[i] = \sum_{j:\beta_j \leq \beta_i} \beta_j\}$
13:      $b[i] \leftarrow N - i$             $\{b[i] = \sum_j \mathbb{1}\,[\beta_j > \beta_i]\}$
14: **end for**
15: **for** $i = 1, 2, \ldots, N$ **do**
16:      $\beta_i \leftarrow S_s[i]$
17:      $a[i] \leftarrow a_{aux}[i]/\beta_i$           $\{a[i] = \sum_{j:\beta_j \leq \beta_i} \beta_j/\beta_i\}$
18:      $AR[i] \leftarrow (a[i] + b[i])/N$
19: **end for**
20: **return** $AR$ and $S_s$        $\{S_s[i]$ is the $\beta$ at which $AR_\beta = AR[i]\}$

---

## G   Further Comparisons with MCMC

Table 4 shows results complementing those of Table 1. RWMH-base and RWMH-R-base denote variants of RWMH and RWMH-R that do not use a mixture distribution to inform the local proposal distribution. Moreover, Table 3 shows results for similar experiments performed on sampling from the "debiased scientist biographies" EBM described in Sec. 3.2.2. Here, there is no obvious way to inform the proposal and therefore, we only use the -base variants for the RWMH sampler. Furthermore, Tables 5, 6, 7, 8 and 9 show a collection of the first 10 samples produced by the QRS, RWMH, RWMH-R, IMH and IMH-R samplers, respectively.

## H   Computing Divergences for Two Poissons

Given Poisson distributions $p$ and $q$ with rates $\lambda_p > \lambda_q > 0$, we wish to find the divergence

$$D_f(p, p_\beta) = \sum_{x=0}^{\infty} \frac{P_\beta(x)}{Z_\beta} f\left(\frac{p(x)Z_\beta}{P_\beta(x)}\right). \tag{37}$$

| sampler | AR | %amazing | PPL↓ | Self-BLEU-5↓ | %Uniq↑ | Dist-1↑ | Dist-2↑ | Dist-3↑ | TVD$(p, p_\beta)$ ↓* | $D_{\mathrm{KL}}(p, p_\beta)$ ↓* |
|---|---|---|---|---|---|---|---|---|---|---|
| DPG | 1 | $62.9 \pm 0.4$ | $61.7 \pm 0.3$ | $85.8 \pm 0.1$ | $100.0 \pm 0$ | $89.3 \pm 0.1$ | $96.1 \pm 0.0$ | $94.1 \pm 0.0$ | $0.67 \pm 0.00095$ | $1.91 \pm 0.04$ |
| IMH | $10^{-1}$ | $100.0 \pm 0$ | - | $92.0 \pm 0.7$ | $66.5 \pm 3.9$ | $89.9 \pm 0.3$ | $96.3 \pm 0.1$ | $94.3 \pm 0.1$ | Unk. | Unk. |
| IMH-R | $10^{-1}$ | $100.0 \pm 0$ | $60.8 \pm 1.4$ | $87.1 \pm 0.2$ | $100.0 \pm 0$ | $89.8 \pm 0.2$ | $96.4 \pm 0.1$ | $94.4 \pm 0.1$ | Unk. | Unk. |
| QRS | $10^{-1}$ | $100.0 \pm 0$ | $61.8 \pm 0.8$ | $86.8 \pm 0.4$ | $100.0 \pm 0$ | $89.9 \pm 0.2$ | $96.4 \pm 0.1$ | $94.3 \pm 0.1$ | $0.27 \pm 0.0054$ | $0.45 \pm 0.045$ |
| RWMH-base | $10^{-1}$ | $80.0 \pm 40.0$ | - | $99.6 \pm 0.9$ | $21.8 \pm 28.8$ | $85.8 \pm 7.6$ | $91.1 \pm 11.2$ | $87.2 \pm 14.6$ | Unk. | Unk. |
| RWMH-R-base | $10^{-1}$ | $63.2 \pm 1.3$ | $61.9 \pm 1.2$ | $86.5 \pm 0.3$ | $100.0 \pm 0$ | $88.5 \pm 0.2$ | $96.2 \pm 0.1$ | $94.3 \pm 0.1$ | Unk. | Unk. |
| IMH | $10^{-3}$ | $100.0 \pm 0$ | - | $86.9 \pm 0.3$ | $98.7 \pm 0.5$ | $90.0 \pm 0.2$ | $96.3 \pm 0.1$ | $94.2 \pm 0.1$ | Unk. | Unk. |
| IMH-R | $10^{-3}$ | $100.0 \pm 0$ | $63.4 \pm 1.5$ | $86.7 \pm 0.1$ | $100.0 \pm 0$ | $89.9 \pm 0.2$ | $96.3 \pm 0.1$ | $94.3 \pm 0.1$ | Unk. | Unk. |
| QRS | $10^{-3}$ | $100.0 \pm 0$ | $62.8 \pm 1.6$ | $86.6 \pm 0.2$ | $100.0 \pm 0$ | $90.0 \pm 0.2$ | $96.3 \pm 0.1$ | $94.3 \pm 0.0$ | $0.01 \pm 0.0067$ | $0.011 \pm 0.0093$ |
| RWMH | $10^{-3}$ | $100.0 \pm 0$ | - | $99.8 \pm 0.1$ | $41.0 \pm 35.2$ | $71.5 \pm 36.0$ | $69.4 \pm 33.7$ | $63.1 \pm 31.1$ | Unk. | Unk. |
| RWMH-R | $10^{-3}$ | $100.0 \pm 0$ | $57.7 \pm 3.9$ | $87.8 \pm 0.1$ | $100.0 \pm 0$ | $82.9 \pm 0.4$ | $93.2 \pm 0.4$ | $92.4 \pm 0.3$ | Unk. | Unk. |

Table 4: Further comparisons of samplers on an EBM with a pointwise constraint to include the word "amazing" in the sequence. We do not compute perplexity for IMH and RWMH without reset as it does not yield i.i.d. samples. As noted, TVD and KL for MCMC methods are unknown (i.e. we have no way of estimating them). Where available, we show mean ± one standard deviation over 10 runs. *TVD and KL are estimated on independent sets of $10^6$ samples.

---

QRS samples at $10^{-3}$ acceptance rate

- Cowgirl 87 Care Commission⏎⏎Header now available on our VIP web site.⏎⏎CowboyBillIssues AVAILABLE NOW...⏎⏎Start your website bio examining this amazing cam-
- take note. No wonder Vox isn't having an amazing day.⏎⏎Screen Snipe Dave ends Neo by christening it #DroppedSummer⏎⏎[] @cloudp
- I got an unprecedented amount of reviews years ago because of such an interesting movie that seemed Nude World will be released in some theaters soon. The movie itself was pretty amazing. I only had to hear
- Before Charlie Hebdo rendered its amazing save for lost lives, scholarship was blind to the brutal irony. Gregory Wade accidentally shot three bystanders, at least ten more wounded, and four more dead within three days of
- 4279 watts has passed the bar⏎⏎You can also check your save scores by clicking 'Save Data to PWSW Audience'.⏎⏎Shopping⏎⏎Welters also have amazing customer
- Sculpt your beard to stay stylish when you struggle to look like a nerd. This Collection makes all your lengthening amazing, so show it off - Ruler of Salt, Cone of Salt at Holl
- If you've contacted our sponsor, they know who you are and how hard you work. Unless months pass without a postcard. Best Served to meet everything you demand. You are amazing and are
- I'm sure a lot of people are most excited to see an F-35 Lightning II V2 today because of the amazing kick that's (understandably) in the engine room now that Benson
- Catherine is serviceable, happy, and successful. She loves teaching, having children, and saving time. She regularly works with can and is an amazing listener.⏎⏎Our community is full of
- The bellies of the runs was amazing. A ton of complements were good, including some neighboring dressing room appearances. So long worked up info, my 2014 Mustang had lots of special opening-specific

Table 5: The first 10 examples produced in the first run of QRS on the EBM filtering GPT-2 sequences containing "amazing".

---

RWMH samples at $10^{-3}$ acceptance rate

- The OnePlus 2 is one of the most amazing smartphones we've ever tried. It's an extremely powerful smartphone you've never heard of before, and carries an amazing battery life, or battery life I'm
- The OnePlus 2 is one of the most amazing smartphones we've ever tried. It's an extremely powerful smartphone you've never heard of before, and carries an amazing battery life, or battery life I'm
- The OnePlus 2 is one of the most amazing smartphones we've ever tried. It's an extremely powerful smartphone you've never heard of before, and carries an amazing battery life, or battery life I'm
- The OnePlus 2 is one of the most amazing smartphones we've ever tried. It's an extremely powerful smartphone you've never heard of before, and carries an amazing battery life, or battery life I'm
- The OnePlus 2 is one of the most amazing smartphones we've ever tried. It's an extremely powerful smartphone you've never heard of before, and carries an amazing battery life, or battery life I'm
- The OnePlus 2 is one of the most amazing smartphones we've ever tried. It's an extremely powerful smartphone you've never heard of before, and carries an amazing battery life, or battery life I'm
- The OnePlus 2 is one of the most amazing smartphones we've ever tried. It's an extremely powerful smartphone you've never heard of before, and carries an amazing battery life, or battery life I'm
- The OnePlus 2 is one of the most amazing smartphones we've ever tried. It's an extremely powerful smartphone you've never heard of before, and carries an amazing battery life, or battery life I'm
- The OnePlus 2 is one of the most amazing smartphones we've ever tried. It's an extremely powerful smartphone you've never heard of before, and carries an amazing battery life, or battery life I'm
- The OnePlus 2 is one of the most amazing smartphones we've ever tried. It's an extremely powerful smartphone you've never heard of before, and carries an amazing battery life, or battery life I'm

Table 6: The first 10 examples produced in the first run of RWMH on the EBM filtering GPT-2 sequences containing "amazing".

---

RWMH-R samples at $10^{-3}$ acceptance rate

---

- yay!!! this game is amazing!!!!! a cool physical sci fi action game where you play a hardened junkie with an old girlfriend who's tough
- in the immediate aftermath of the church shutdown, strange and often amazing events raised the question of whether the government had ever given up its restrictive control of religious provocations for political purposes and assumed a new political leadership
- ... had an amazing 7 out of 10, 8 out of 10, 9 out of 10, 6 out of 5... liked it... the story...
- I do have several solo masters who still have done amazing work to suit/complement orientated-character traits. In particular, I find myself building out secondary mazes based on their chunks iay
- "She and my son played "Friends," and used songs that I loved from that's were amazing," Kleutz said. "They were acknowledging responsible themes. They felt I was so passionate that I
- useful for those in need or those who are distracted in the outside world. qualms ( # 77574 ), pro / pro bizarre ( # 776171 ), amazing 1 / 2 / 3 / 5 ( # 776249 )
- We are inspired by the amazing work of the Reader Space Foundation, which fosters artists who make an impact on a global audience through knowledge, support, and advocacy. It's made us an un-
- the casanova is amazing the key to this being a classic, a true black lace bra made from dl's, lace and all the fabrics that are available in pink and black. go on, have some fruit, sugar, vitamins, icing. whisper, wow, good for you
- and it is only using standard rules of thumb, but it is amazing that all the use of " " can be taken into account. " " " " " " " " " " " " " " " " " " " " " " " " " established.
- With so many amazing actors appearing on every level in a stone once it's filmed, Star Wars Rebels follows the Rebellion throughout 2017 with an unusual archive of credits from the original and Rebels Two Casts.

---

Table 7: The first 10 examples produced in the first run of RWMH-R on the EBM filtering GPT-2 sequences containing "amazing".

---

IMH samples at $10^{-3}$ acceptance rate

---

- OF ALL The Verifiers who voted for Dragon Quest Revealed, two amazing choices are going to still be in the fire. To make up for the lost depth, Vulcans will make a busy
- A PO3 sniff tested, protecting the lab from detection by probiotic products, is amazing.⏎⏎5 years ago, scientists found the cannabinoid-blocking drug O3 responsible
- You have watched The Witcher 3 as a kid. You have eaten plenty of meat at your meals. You've seen a video game show before that has been amazing. Very few people have grown up without
- 11 am 2pm 958-701-99997 to Lift EcoTV, located in Washington, D.C. Do you have more questions about lifting something amazing in the Farm 600 in⏎
- Bringing this gift is a unique opportunity to share with your loved ones about the home we love. Our family stunts cloud of feedback and the truly amazing handmade items we came upon included, providing a great
- Over a group of homebrew enthusiasts gathered for a beer sampling on Wednesday morning, they shared some amazing homebrew recipes with beer enthusiasts on the craft beer of the weekend, as well as some star brewers at home
- RIDER VIDEOTAPE | LEON BEVERLY RON OLD AFAIK proud to present you the amazing RADIO available from RON Old Hollywood: http://www.personalradio
- with comments, screenshots⏎⏎collaborating authors & planners⏎⏎Connecting friends & family, live with family of our wonderful amazing people.⏎⏎An Albany Folklore Center where we are based
- Could She Play The Same Sex Career?⏎⏎Missoula is full of different, amazing lesbian and gay couples to come. It's so amazing that she's so welcoming to them. They love
- amazing even or nerdyally⏎⏎I mean it's true that almost every person touching your skin is under their own skin, but when you have to contribute, you are competing against art first

---

Table 8: The first 10 examples produced in the first run of IMH on the EBM filtering GPT-2 sequences containing "amazing".

---

IMH-R samples at $10^{-3}$ acceptance rate

---

- Journey to Mars! Supermassive Black Hole Recovered From Big Bang Rosetta Sky⏎⏎It took an amazingly severe situation to produce an amazing HDR image through single recombination with massive radio uplink
- Early in the credits game, Sid Meier looked over Randi with his smugly excited face.⏎⏎"What if," he started, "if you think Randi's doing amazing things to me
- While Gilinda tells everybody what she has in mind in the past, notice that despite their competitiveness and amazing athleticism, they do outlive the Hokies and don't recognize one another.⏎⏎⏎It
- ameral bairnsaucer, from Prague⏎⏎No, this wasn't a blast. It was awesome. My test team was amazing.⏎⏎If you wait against an scent that is
- Last day was so amazing. I was going to lay down for some lunch on the park deck and watch contestants go nuts. Happy morning.⏎⏎Psychos Solid: Winning the Super Bowl isn't
- Can Customers Generation Support Field Trawler?⏎⏎Well, it looks like it's gonna be well taken care of...Tousting can also apparently utilise compiled sound block and got some amazing gimm
- 64 Explicit Nov 16, 2016 FIVE US SECS CONCLUSIONS On Saturday night you'll find an amazing bunch of colleges and universities alike as they host the 'THE LAST TENSE' panel for
- Pick anything you wish for appears in new like Black Bag Revenge, Savage Comet, Bride of Waibou, Oinnibox and much more.⏎⏎Gamazing Of The Month Star Trade Reply
- At 6 miles in, the Gawdover Range is an amazingly located range within Markham county, NY. I was able to flag down one during the last few kilometres and here is the fun!
- I'm never happy about an OC for a while, since I can like them because the fact that the beautiful side has proven amazing makes sure I learned to love them more and remain satisfied when looking at

---

Table 9: The first 10 examples produced in the first run of IMH-R on the EBM filtering GPT-2 sequences containing "amazing".

As $(\lambda_p/\lambda_q)^x$ is increasing in $x$, for any fixed $\beta > 0$ and nonnegative integer $x$ we have

$$p(x) \le \beta q(x) \qquad \Leftrightarrow \qquad (\lambda_p/\lambda_q)^x \le \beta \exp(\lambda_p - \lambda_q) \qquad \Leftrightarrow \qquad x \le k$$

where

$$k \doteq \left\lfloor \frac{\lambda_p - \lambda_q + \log \beta}{\log(\lambda_p/\lambda_q)} \right\rfloor.$$

Thus the partition function appearing in Eq. 37 is

$$Z_\beta = \sum_{i=0}^{\infty} \min\left(p(i), \beta q(i)\right) = \sum_{i=0}^{k} p(i) + \beta \sum_{i=k+1}^{\infty} q(i) = F_{\lambda_p}(k) + \beta(1 - F_{\lambda_q}(k)),$$

where $F_\lambda$ is the cumulative distribution function of a Poisson of rate $\lambda$.

Given this value of $Z_\beta$, the $f$-divergence in Eq. 37 may now be approximated by truncating it to $n$ terms for a suitably large $n$, provided the convex function $f$ does not grow too rapidly:

$$D_f(p, p_\beta) \approx \sum_{x=0}^{n} \frac{P_\beta(x)}{Z_\beta} f\left(\frac{p(x) Z_\beta}{P_\beta(x)}\right).$$

## I  Importance Sampling and Variance

In this work, we perform many IS estimates of quantities such as acceptance rate, TVD and KL divergence. We report variance estimates computed using the bootstrap estimator for the experiments on debiasing scientist biographies (Sec. 3.2.2) and other EBMs reported in Appendix A in Table 10. We report mean $\pm$ one standard deviation for $\beta$ values within the range used in our experiments (also see Table 2). We find our estimates to be accurate within reasonable variance.

Secondly, as noted in Sec. 2.2, it is not generally possible to compute $Z$ or its variance for any given EBM with certainty. This observation can put into question the validity of the above-described estimates. We note, however, that when we can bound $P(x)/q(x)$ we can have formal bounds on these quantities, allowing us to double-check the accuracy of these estimates. In particular, this is possible when $P(x) = a(x)b(x)$, $q(x) = a(x)$ and $b(x) \in [0, M]$, as in that case we have $P(x)/q(x) \le M$. Using Popoviciu's inequality (Popoviciu, 1935), we know that

$$\text{Var}\left[\frac{P(x)}{q(x)}\right] \le \frac{M^2}{4}, \tag{38}$$

and thus, the IS estimator of $Z$ has variance

$$\text{Var}\left[\frac{1}{N} \sum_{i=0}^{N} \frac{P(x_i)}{q(x_i)}\right] \le \frac{M^2}{4N}. \tag{39}$$

We generated 1M samples with GPT-2, and used them to compute the partition functions of the EBMs for pointwise constraints, with provable bounds on their variance following Eq. 39. For the 100% "amazing" EBM, we obtained $Z = 2.5 \times 10^{-3} \pm 2.5 \times 10^{-7}$; whereas for 100% "wikileaks" we obtained $Z = 1.5 \times 10^{-4} \pm 2.5 \times 10^{-7}$. These are in agreement with the estimates in Table 10. However for our experiments with distributional constraints, we determinined the rather large bound $M = 304868$, and we would need to gather at least $10^{12}$ samples to obtain a reasonable bound on the variance. This highlights the need for better proposal distributions to compute these quantities, especially considering that the number of samples that IS needs to compute partition function of $P(x)$ is inversely proportional to $\exp(D_{\text{KL}}(p, q))$ (Chatterjee and Diaconis, 2018). This is the reason why, *in practice*, a good proposal distribution can allow us to compute accurate estimates, even if that comes at the loss of the above-described formal bounds.

| $P$ | $\beta$ | AR | $\mathrm{TVD}(p, p_\beta)$ | $D_{\mathrm{KL}}(p, p_\beta)$ |
|---|---|---|---|---|
| 100% amazing | $0.02$ | $0.08 \pm 1.8 \times 10^{-4}$ | $0.2 \pm 6.5 \times 10^{-3}$ | $0.3 \pm 0.04$ |
| | $0.3$ | $7.8 \times 10^{-3} \pm 3.2 \times 10^{-5}$ | $0.04 \pm 8.0 \times 10^{-3}$ | $0.05 \pm 0.02$ |
| | $2$ | $1.2 \times 10^{-3} \pm 6.8 \times 10^{-6}$ | $0.01 \pm 6.7 \times 10^{-3}$ | $0.01 \pm 9.3 \times 10^{-3}$ |
| 100% wikileaks | $7.4 \times 10^{-4}$ | $0.1 \pm 2.1 \times 10^{-4}$ | $0.3 \pm 5.2 \times 10^{-3}$ | $0.6 \pm 0.03$ |
| | $0.01$ | $8.5 \times 10^{-3} \pm 3.7 \times 10^{-5}$ | $0.07 \pm 7.3 \times 10^{-3}$ | $0.09 \pm 0.02$ |
| | $0.2$ | $8.2 \times 10^{-4} \pm 6.6 \times 10^{-6}$ | $9.7 \times 10^{-3} \pm 3.5 \times 10^{-3}$ | $3.9 \times 10^{-3} \pm 1.9 \times 10^{-3}$ |
| 50% female | $1 \times 10^{1}$ | $0.1 \pm 1.3 \times 10^{-4}$ | $0.02 \pm 6.2 \times 10^{-4}$ | $0.01 \pm 1.2 \times 10^{-3}$ |
| | $1 \times 10^{2}$ | $0.01 \pm 1.5 \times 10^{-5}$ | $5.9 \times 10^{-4} \pm 2.9 \times 10^{-4}$ | $3.3 \times 10^{-4} \pm 1.8 \times 10^{-4}$ |
| | $2 \times 10^{3}$ | $1.1 \times 10^{-3} \pm 1.5 \times 10^{-6}$ | $2.7 \times 10^{-7} \pm 4.4 \times 10^{-7}$ | $-1.7 \times 10^{-8} \pm 9.4 \times 10^{-7}$ |
| 50% female + 100% science | $7 \times 10^{3}$ | $0.08 \pm 1.7 \times 10^{-4}$ | $0.3 \pm 6.0 \times 10^{-3}$ | $0.5 \pm 0.03$ |
| | $1.1 \times 10^{5}$ | $7.1 \times 10^{-3} \pm 3.5 \times 10^{-5}$ | $0.07 \pm 7.1 \times 10^{-3}$ | $0.08 \pm 0.02$ |
| | $6.4 \times 10^{5}$ | $1.3 \times 10^{-3} \pm 1.0 \times 10^{-5}$ | $0.02 \pm 4.9 \times 10^{-3}$ | $9.9 \times 10^{-3} \pm 4.8 \times 10^{-3}$ |
| 50% female + 100% sports | $1.2 \times 10^{5}$ | $0.07 \pm 1.3 \times 10^{-4}$ | $0.1 \pm 4.2 \times 10^{-3}$ | $0.2 \pm 0.01$ |
| | $7.5 \times 10^{5}$ | $0.01 \pm 3.5 \times 10^{-5}$ | $0.04 \pm 3.8 \times 10^{-3}$ | $0.04 \pm 6.2 \times 10^{-3}$ |
| | $1.2 \times 10^{7}$ | $8.5 \times 10^{-4} \pm 4.6 \times 10^{-6}$ | $4.2 \times 10^{-4} \pm 3.2 \times 10^{-4}$ | $4.2 \times 10^{-5} \pm 3.6 \times 10^{-5}$ |

| $P$ | $\beta$ | $Z$ | $Z_\beta$ |
|---|---|---|---|
| 100% amazing | $0.02$ | | $0.08 \pm 1.8 \times 10^{-4}$ |
| | $0.3$ | $2.5 \times 10^{-3} \pm 2.5 \times 10^{-5}$ | $7.8 \times 10^{-3} \pm 3.2 \times 10^{-5}$ |
| | $2$ | | $1.2 \times 10^{-3} \pm 6.8 \times 10^{-6}$ |
| 100% wikileaks | $7.4 \times 10^{-4}$ | | $0.1 \pm 2.1 \times 10^{-4}$ |
| | $0.01$ | $1.4 \times 10^{-4} \pm 1.4 \times 10^{-6}$ | $8.5 \times 10^{-3} \pm 3.7 \times 10^{-5}$ |
| | $0.2$ | | $8.2 \times 10^{-4} \pm 6.6 \times 10^{-6}$ |
| 50% female | $1 \times 10^{1}$ | | $0.1 \pm 1.3 \times 10^{-4}$ |
| | $1 \times 10^{2}$ | $2 \pm 2.3 \times 10^{-3}$ | $0.01 \pm 1.5 \times 10^{-5}$ |
| | $2 \times 10^{3}$ | | $1.1 \times 10^{-3} \pm 1.5 \times 10^{-6}$ |
| 50% female + 100% science | $7 \times 10^{3}$ | | $0.08 \pm 1.7 \times 10^{-4}$ |
| | $1.1 \times 10^{5}$ | $8 \times 10^{2} \pm 9$ | $7.1 \times 10^{-3} \pm 3.5 \times 10^{-5}$ |
| | $6.4 \times 10^{5}$ | | $1.3 \times 10^{-3} \pm 1.0 \times 10^{-5}$ |
| 50% female + 100% sports | $1.2 \times 10^{5}$ | | $0.07 \pm 1.3 \times 10^{-4}$ |
| | $7.5 \times 10^{5}$ | $1 \times 10^{4} \pm 6 \times 10^{1}$ | $0.01 \pm 3.5 \times 10^{-5}$ |
| | $1.2 \times 10^{7}$ | | $8.5 \times 10^{-4} \pm 4.6 \times 10^{-6}$ |

Table 10: Means and standard deviation of IS estimates of acceptance rate, TVD with the target distribution and KL divergence to the target distribution for various $\beta$ on various EBMs using a DPG fine-tuned proposal. We perform 5,000 bootstrap simulations using 1,000,000 samples each to compute the means and standard deviations. Values of $\beta$ are chosen within the range used for our experiments as reported in Table 2.

**Estimating $Z$: an example with finite mean and infinite variance**   As mentioned in footnote 6, the variance $\text{Var}_{x \sim q} \frac{P(x)}{q(x)}$ may be infinite. For instance, if $P$ and $q$ are distributions satisfying $P(x)^2 = q(x)$ on support $x = 0, 1, \ldots$, then

$$\mathbb{E}_{x \sim q} \frac{P(x)^2}{q(x)^2} = \sum_{x=0}^{\infty} \frac{P(x)^2}{q(x)} = \sum_{x=0}^{\infty} 1 = \infty.$$

As the variance is the mean-square minus the squared-mean, it too is infinite. The condition $P^2 = q$ holds for appropriately normalized geometric distributions $p(x) \propto a^x$ and $q(x) \propto a^{2x}$ for any $0 < a < 1$.

## J   About the Metropolis-Hastings Algorithm

Robert and Casella (2004, Theorem 7.4, p. 274) prove the following theorem, with $f$ the target distribution. Our motivation for reproducing this result is to emphasize point (ii), which is especially relevant in the context of this paper.

**Theorem J.1.** *Suppose that the Metropolis-Hastings Markov chain $(X^{(t)})$ is $f$-irreducible.*

*(i) If $h$ is an $f$-integrable function, then*

$$\lim_{T \to \infty} \frac{1}{T} \sum_{t=1}^{T} h(X^{(t)}) = \int h(x) f(x) \, \mathrm{d}x \qquad \textit{almost surely.}$$

*(ii) If, in addition, $(X^{(t)})$ is aperiodic, then*

$$\lim_{n \to \infty} \text{TVD} \left( f, \int K^n(x, \cdot) \, \mu(\mathrm{d}x) \right) = 0$$

*for every initial distribution $\mu$, where $K^n(x, \cdot)$ denotes the kernel for $n$ transitions.*

Property (i) says that the average over the $T$ first elements of a *single* chain converges to the expectation of $h(x)$ for $x \sim f$, as $T$ increases. Property (ii) is concerned with the TVD between the target distribution $f$ and the distribution obtained by repeatedly running an $n$-step chain and outputting the $n^{\text{th}}$ element — the *reset* (-R) variant of the MH algorithm that we have denoted by RWMH-R or IMH-R. This distance converges to zero as $n$ increases.

## K   Arbitrarily Slow Proposal $q$

We assume that $X = \mathbb{N} = \{0, 1, 2, \ldots\}$, that $p$ has full support and is strictly decreasing over $\mathbb{N}$.[18]

**Theorem K.1.** *Let $g : (0, \infty) \to [0, 1]$ be a strictly decreasing continuous function with $\lim_{\beta \to \infty} g(\beta) = 0$. Then there exist a real number $\beta_0 > 0$ and a proposal probability distribution $q$ over $\mathbb{N}$ such that the QRS distribution $p_\beta$ for target $p$ satisfies*

$$\text{TVD}(p, p_\beta) \geq g(\beta) \qquad \textit{for all } \beta > \beta_0. \tag{40}$$

*tribution associated to $p$ and $q$ at $\beta$.*

*Proof.* See below.   □

In other words, whatever the "slowness" with which $g(\beta)$ decreases to 0, we can always find a proposal $q$ that guarantees that, asymptotically, $\text{TVD}(p, p_\beta)$ decreases even more slowly.[19]

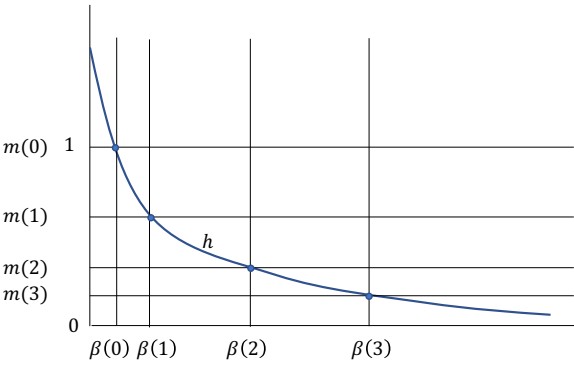

Figure 13: Illustration of Lemma K.2.

## Proof sketch

The proof, which we only sketch, is based on two lemmas.

**Lemma K.2.** *If $q$ is a distribution over $\mathbb{N}$ with full support, let $V_\beta \doteq \{x \in \mathbb{N} : p(x)/q(x) \geq \beta\}$. Let $h(\beta)$ be any strictly decreasing continuous function from $(0, \infty)$ to $[0, 1]$, such that $\lim_{\beta \to \infty} h(\beta) = 0$. Then there exist some $q$ and $\beta_1 > 0$ such that*

$$\beta \geq \beta_1 \qquad \Rightarrow \qquad p(V_\beta) \geq h(\beta).$$

*Proof.* For $x \in \mathbb{N}$, let $m(x) \doteq p(\{x, x+1, \ldots\})$ and $\beta(x) \doteq \inf\{\beta : h(\beta) \leq m(x)\}$. This is easily seen to imply that $h(\beta(x)) = m(x)$ and that $\beta(x)$ is strictly increasing (see Fig. 13). Then construct a distribution $q$ over $\{0, 1, 2, \ldots\}$ such that for some $x_0 \geq 1$, and for any $x \geq x_0$, one has $q(x) \doteq \frac{p(x)}{\beta(x+1)}$; and we set $\beta_1 \doteq \beta(x_0)$. We note that the intervals $[\beta(x_0), \beta(x_0+1)), [\beta(x_0+1), \beta(x_0+2)), \ldots$ cover the whole interval $[x_0, \infty)$. Then consider some integer $k \geq x_0$ and take $\beta$ in the interval $[\beta(k), \beta(k+1))$. For $x \geq k$, we have $\frac{p(x)}{q(x)} = \beta(x+1) \geq \beta(k+1) \geq \beta$, hence $x \in V_\beta$. But $p(\{k, k+1, \ldots\}) = h(\beta(k)) \geq h(\beta)$, hence $p(V_\beta) \geq h(\beta)$. $\qquad \square$

**Lemma K.3.** *Let $q$ be a distribution over $\mathbb{N}$ with full support, and let $W_\beta \doteq \{x \in \mathbb{N} : p(x)/q(x) \geq 3\beta\}$. Then there exists a real number $\beta_2 > 0$ such that*

$$\beta \geq \beta_2 \qquad \Rightarrow \qquad \text{TVD}(p, p_\beta) \geq p(W_\beta)/2.$$

*Proof.* We have

$$p_\beta(x) = \frac{p(x) \wedge (\beta q(x))}{Z_\beta} = \frac{p(x)}{Z_\beta} \left( 1 \wedge \left( \beta \frac{q(x)}{p(x)} \right) \right), \qquad Z_\beta = \sum_x p(x) \wedge (\beta q(x)).$$

Hence, for $x \in W_\beta$,

$$\frac{p_\beta(x)}{p(x)} = \frac{1 \wedge \left( \beta \frac{q(x)}{p(x)} \right)}{Z_\beta} \leq \frac{1}{3 Z_\beta}.$$

Let $\beta_2 > 0$ be such that $\beta \geq \beta_2 \Rightarrow Z_\beta > 0.9$. (It is easy to see that such a $\beta_2$ exists, as $Z_\beta$ is increasing in $\beta$ and converges to 1.) Then for $\beta \geq \beta_2$ and for $x \in W_\beta$, we have $\frac{p_\beta(x)}{p(x)} \leq \frac{1}{3 \times 0.9} \leq \frac{1}{2}$, so that

$$p(x) - p_\beta(x) = p(x) \left( 1 - \frac{p_\beta(x)}{p(x)} \right) \geq \frac{p(x)}{2},$$

and therefore $\text{TVD}(p, p_\beta) = \sum_{x : p(x) \geq p_\beta(x)} (p(x) - p_\beta(x)) \geq p(W_\beta)/2$. $\qquad \square$

---

[18]These conditions are WLOG, as can easily be checked.
[19]Based on the properties proved in the paper, we know that eventually $\text{TVD}(p, p_\beta)$ decreases to 0.

*Proof of Theorem K.1.* From these two lemmas, the proof of the theorem follows easily. Indeed, let $h(\beta') \doteq \min(1, 2\,g(\beta'/3))$. For $\beta' = 3\,\beta$, we have $W_\beta = V_{\beta'}$, and therefore, by Lemma K.3, $\mathrm{TVD}(p, p_\beta) \geq p(V_{\beta'})/2$, for $\beta \geq \beta_2$. By Lemma K.2, we have $p(V_{\beta'}) \geq h(\beta') = 2g(\beta)$ for $\beta' \geq \beta'_1$, so that $\mathrm{TVD}(p, p_\beta) \geq g(\beta)$ for $\beta$ large enough that $g(\beta) \leq 1/2$, and we conclude. $\qquad\square$

