# OpenReview forum: "An approximate sampler for energy-based models with divergence diagnostics"
_TMLR — Accepted by TMLR_

### Review · Reviewer_6dL8 · 2022-09-14

**Summary Of Contributions:**

The authors study quasi-rejection sampling (QRS) for discrete distributions. QRS is akin to rejection sampling (RS), except that the scaled proposal may be improper, i.e., may not be a global upper bound on the unnormalized target distribution. This means that unlike RS, QRS is not an exact sampler. Nevertheless, the authors motivate the use of QRS because it leads to higher acceptance ratios. By tuning the value of a parameter β which controls the degree to which the scaled proposal is improper (letting β tend to infinity, one recovers RS), one can achieve a trade-off between the computational complexity of the algorithm (related to the acceptance ratio) and the quality of the sample (how close it is to the target).

The other motivation for QRS is that unlike Markov chain Monte Carlo (MCMC), by using importance sampling estimates together with the explicit proposal, one can obtain estimates on the f-divergence of the output from the true target distribution. This addresses the important practical problem of obtaining convergence diagnostics for sampling.

Although there is no theoretical guarantee for the accuracy of the sample as a function of β, the authors do prove the non-trivial fact that the f-divergence of the output of the sample to the target decreases monotonically as a function of β. The approach is also validated with experiments on real-world data.

**Broader Impact Concerns:**

None.

**Requested Changes:**

Please discuss the limitations of RS and importance sampling further. Are there simple assumptions which entail that the TVD between p_β and p goes down as an explicit function of β, e.g., at rate 1/β?

For the definition of the f-divergence in eq. (1), this is only valid if p_1 is absolutely continuous with respect to p_2, or if f grows superlinearly. In the case when f grows linearly at infinity, which is the case for the TV distance, one needs to add an extra term consisting of f’(∞) times the measure under p_1 of the set {p_2 = 0}. (In the current definition, one can see that if p_1 and p_2 have disjoint supports and the f-divergence is the TV distance, the formula (1) yields 0 but the true TV distance is 1.)


**Strengths And Weaknesses:**

I found the theoretical component of the paper to be thin, as there is essentially no quantitative justification for the procedure. I do find the theorem on monotonicity of the f-divergence as a function of β to be interesting. The current proof, based on the spread lemma for convex functions, is mysterious to me. I propose an alternative proof based on calculus (the proof assumes that f is differentiable, but I believe that this restriction can be lifted via approximation arguments), see https://drive.google.com/file/d/17ZFUnGBdzhDnC6r6WEPIznRX68N2YzAI/view?usp=sharing.

Although the use of QRS is motivated by the importance sampling estimators for the f-divergence, in general I believe that the variance of these estimators typically scales exponentially with the size of the state space. This to me seems like a serious disadvantage of the proposed algorithm. Also, since QRS is simply an approximate form of RS, it shares its shortcomings. Namely, the acceptance probability of RS is often too low to be practical; although QRS can increase the acceptance probability by setting β to be small, it does so at the cost of sacrificing sample quality (and this is not quantified well in this work). To counter this point, the authors claim that neural networks can provide good global proposals for RS and QRS, but this is rather handwavy and unsubstantiated.

I find the experiments to be a strength of this paper, since they demonstrate well that in certain applications QRS can be effective. Also, I’d like to add that the paper is extremely well-written. My main doubts lie in whether QRS is useful more generally.

Overall, although QRS is fairly natural, I feel that this paper could benefit from further investigation. For example, perhaps the authors could further investigate QRS in some simple test settings in order to investigate more quantitatively the trade-off for the value of β, the variance of the importance sampling estimators, and comparison with mixing times of natural Markov chain algorithms. Besides this point, I feel that some of the limitations (e.g., that non-adaptive RS is typically not used beyond simple settings) are not discussed well.

---

> ### Author Response · Authors · 2022-10-21
> **Response to Reviewer 6dL8 Part 1/2**
>
> Thank you for your kind words about a number of aspects of our work. We will try to answer the more critical parts of your review and your suggestions for improvements.
>
> First of all, thank you for taking the time to provide an alternative proof for our monotonicity result (Theorem 2.2). Actually, at an earlier stage of our work, we attempted to provide a calculus-based proof in a similar spirit to what you are proposing. We were however concerned about the non-differentiability aspects, *which we believe are serious and need to be confronted head-on*, due both to possible non-differentiability of $f$, but also to the non-differentiability of the *min* function, made even worse by the fact that the points $\beta$ at which this *min* is computed can be infinitely many inside a finite interval (therefore, with some accumulation points). So, we decided to go for a more elementary proof, not requiring differentiability, and also providing some geometric intuitions related to "Jensen gaps". Admittedly, the currently provided proof is rather long and dense, so based on your suggestion, we have added in the Appendix D.4 a complete version of our initial "calculus-based" attempt which exploits an extension of the fundamental theorem of calculus not requiring differentiability, but only absolute continuity (which follows from Lipschitz continuity). As a side remark, please note that we believe there is
> a typo in the proof that you provided; On p. 2, you say "Towards this end, first we claim that $1/q(A_\beta) \leq \beta/Z_\beta$; This is equivalent to $Z_\beta \leq \beta q(A_\beta)$", while what you show later in the sentence actually corresponds to the reverse inequalities.
>
> In any case, thanks again a lot for your great suggestion, and we hope our additional proof will satisfy you.
>
> The main criticism raised by the review in different places is the lack of sufficient "quantitative justification" for our procedure, as well as doubts regarding conditions under which it may be reasonably efficient.
>
> For instance:
> *"In general, I believe that the variance of these estimators typically scales exponentially with the size of the state space"*. We are not sure what is the basis for this statement, and which reference would support it. Our state space $X$ is typically infinite, and the empirical evidence of our experiments in App. I ("Importance Sampling and Variance") exploiting bootstrap resampling estimates of the variance indicate not only finiteness but also stability of the variance (see Table 10). In App. I, we also compare situations where the IS concerns a *bounded* R.V., for which one can bound the variance *a priori*, with situations where such an *a priori* guarantee is impossible. As can be verified in standard references about Importance Sampling, such as [1], in practice I.S. is typically used in situations where only empirical estimates of the variance are possible, which does not undermine its great value in numerous applications.
>
> Concerning the lack of "quantitative justification" and the fact that the trade-off between acceptance probability and sampling quality is "not quantified well in this work", or the claim that QRS is not used beyond "simple settings", we have to respectfully, but strongly, disagree. We believe that we already do quite a lot in these respects, for instance in terms of Theorem 2.1 bounding the TVD in terms of an easy to interpret $p$-mass of violators; or in terms of the many empirical estimates of $f$-divergences that we provide. And it is not clear to us in what ways the several settings coming from NLP applications can be said to be simple: they do not appear to us to be so in comparison to existing work in these domains.
> Concerning the statement *"the authors claim that neural networks can provide good global proposals for RS and QRS, but this is rather handwavy and unsubstantiated"*, we also have to disagree. Current neural networks have an impressive ability to approximate complex target distributions (for instance via the recent generic DPG fine-tuning algorithm that we cite), a property that can be exploited to provide global proposals that are a good starting point for QRS, as we show in some of our experiments. Another way to exploit large pretrained neural networks (aka "foundational models") is through fast-evolving "prompt engineering" techniques, which have the potential of being used as approximations to some underlying more formal distributional target, as we briefly sketch in the submission.
>
> [1] Owen 2013 - Unpublished Lecture Notes https://artowen.su.domains/mc/Ch-var-is.pdf
>
> Part 1/2

---

> > ### Author Response · Authors · 2022-10-21
> > **Response to Reviewer 6dL8 Part 2/2**
> >
> > Finally, we would like to stress two distinctive aspects of our setup. First, we work over a discrete countable space $X$, which does not necessarily have any of the structural properties (metric, normed, Euclidian, or even ordered) of the more common continuous spaces, for which it is easier to define functional properties of probability densities and provide mathematical analysis. Second, in practice both the target $p$ and the proposal $q$ incorporate complex trained neural networks for which no closed analytical form is available and on which we have little direct control; in other words, we cannot easily impose  mathematical properties *a priori* on these distributions, which might facilitate analysis, but would limit applications in practice.
> >
> > *Requested changes.*
> >
> > Concerning $f$-divergence, thank you for noting the mistake in our current definition! Indeed, when $p_1 \not\ll p_2$ ($p_1$ not abs. cont. relative to $p_2$), one should add the term you mention. We correct this omission in the updated submission, and indicate that because we assume that $p\ll q$ (see iii.\ at the beginning of Section 2), we have both $p_\beta \ll p$ and $p \ll p_\beta$, ensuring that the rest of our developments do not need this additional term and are valid.
> >
> > Regarding the limitations of RS and IS, we tried to address at least some of these concerns above.
> >
> > Concerning the question "Are there simple assumptions which entail that the TVD between $p_\beta$ and $p$ goes down as an explicit function of $\beta$, e.g., at rate $1/\beta$?", we thank you for raising this point, which we had not considered before. Our attempt at a response consists in an easy and in a more difficult, novel result, spurred by your question (both results assume, as usual, that $p\ll q$). The easy and positive part is to observe that for a *finite* space $X$, there is always a threshold $\beta_0$ such that beyond this threshold, $\text{TVD}(p,p_\beta) = 0$: just take $\beta_0 = \max_{x\in X} p(x)/q(x)$. The more difficult, and negative result, consists in the following theorem, going in the opposite direction, detailed and proven in App. K of the updated submission: for any target distribution $p$ over an infinite countable sample space, and for any arbitrarily slowly decreasing function $g(\beta)$ converging to $0$ in the limit, there exists some proposal $q$ such that $\text{TVD}(p,p_\beta)$ decreases *even more slowly* than $g(\beta)$. In other words, if one is interested in playing the devil's advocate, it is always possible to design an arbitrarily bad proposal in terms of asymptotic convergence rate.
> >
> > Part 2/2

---

> > > ### Comment · Reviewer_6dL8 · 2022-10-22
> > > **Response**
> > >
> > > I thank the authors for their detailed response, and I am very sorry to hear about the family emergency.
> > >
> > > The authors have addressed pretty much all of the concerns I have raised. The only comment I'd like to make is that I still find the claim about neural networks to be handwavy; while I agree that neural networks have spectacular empirical performance on various tasks, our understanding of neural networks is far too little to be able to claim specific remarks such as "neural networks can produce good global proposals" (in what settings? in what sense of "good"? how much data/training is needed?). Nevertheless I am happy to recommend that this work be accepted for publication.

---

### Review · Reviewer_ZGKb · 2022-09-27

**Summary Of Contributions:**

This work proposes a relaxation of rejection sampling, called QRS, that replaces the strict upper bound for the proposal distribution with an approximate upper bound. The use of an approximate upper bound allows the method to be applied in cases when a true upper bound is unknown, and can greatly increases the efficiency of the sampler at the cost of accuracy compared to classic rejection sampling. Moreover, the QRS method can provide approximate divergence diagnostics that allow the user to estimate the sampling error incurred from using an approximate upper bound. The divergence diagnostics provide a way to trade off between accurate approximation of the target distribution and sampling efficiency. The QRS method is applied to sample from language models using a large pre-trained model as an initial proposal distribution. Experiments show that QRS is able to effectively fine-tune language models to produce outputs that are compatible with user-defined criteria such as gender parity and inclusion of specific topics. The divergence diagnostics show that the method is able to strike an effective balance between sampling accuracy and sampling efficiency, with the notable ability to gain speedup of an order of magnitude or more with only slight loss of sampling accuracy.

**Broader Impact Concerns:**

The authors sufficiently address ethical concerns.

**Requested Changes:**

Significant Changes:

* It would be helpful to show the results of a MCMC and IMH for the toy Poisson experiment for baseline comparison vs. QRS.

Minor changes:

* First paragraph of Page 1: "Such models map elements of the sample space to non-negative scores." The "non-negative" part is not true in general, as energy functions can often take on negative values (especially in the context of the 2006 LeCun paper where the energy function is not necessarily probabilistic). The unnormalized density $e^{- E(x)}$ is non-negative, but the energy function $E(x)$ might be negative.
* Algorithm 1: might be helpful to indicate that the algorithm will return the samples $x$ as the final output (maybe change "While True do:" to "While num_samples_generated < num_samples_target")
* Page 4 top: change "contains many discrepancies" to "contains many common divergence objectives"
* Section 3.2.3 Baselines paragraph: typo "focussing" instead of "focusing"

**Strengths And Weaknesses:**

Strengths:

* The proposed method is straightforward and sensible. A major benefit of the proposed method over existing sampling methods is tractability of divergence diagnostics and improved sampling efficiency compared to standard rejection sampling. This comes at the cost of approximate sampling, which can be quantified with divergence diagnostics and which appears to be close to the true distribution in experiments.
* The paper provides nice development of theoretical properties, in particular the result in Theorem 2.2 that increasing $\beta$ strictly improves the approximation of the target distribution.
* Experimental results show solid performance for fine-tuning language models to produce specific outputs. Plots in Figures 3 and 4 show that the divergence diagnostics are useful for measuring the approximate convergence of the sampler across different settings of $\beta$.
* Reasonable comparisons are made between QRS and existing MCMC methods, specifically Random Walk MH and IMH.

Weaknesses:
* The method relies on the existence of a good global proposal distribution, which limits its utility as a general-purpose sampler. However, the language modeling setting provides a strong example of a situation where using an EBM to fine-tune a global proposal provides significant benefits.
* The title is somewhat confusing. It seems to imply that divergence diagnostics are used for sampling, instead of being available after sampling to assess the sampling results. It might be good to mention "approximate rejection sampling" or something similar in the title.
* The baseline IMH approach gives very similar performance to QRS across the metrics investigated. The main benefit of QRS over IMH is the ability to calculate divergence diagnostics. The paper investigates a lower-bound for the TVD that results from sampling with IMH, but this lower bound is nearly identical to the TVD from the QRS diagnostic. Thus, there is no direct evidence in favor of applying QRS instead of IMH. Nonetheless, IMH is still incompatible with easily-obtained divergence diagnostics. A clear case where QRS outperforms IMH in divergence diagnostics would strengthen the paper, although I can still see the appeal of QRS over IMH from the perspective of straightforward diagnostics.

---

> ### Author Response · Authors · 2022-10-21
> **Response to Reviewer ZGKb**
>
> Thank you very much for the  detailed feedback on our work, and for your generally positive assessment.
>
> Concerning the very useful minor changes requested, we have tried to handle all of them in the revision, apart from the one concerning the spelling of "focussing", that we keep in an attempt to stay consistent with British conventions that we have generally adopted in the paper.
>
> We have also changed the title in order to remove the ambiguity that you noted.
>
> Concerning IMH vs. QRS, we agree with you that the experiments in the paper provide no clear evidence of the superiority of QRS over IMH, at least in terms of lower bounds for divergences based on the binning projections that we explored in these experiments, and on the original (we believe) exploitation of the DPI for estimating these lower bounds. However, as you note, the QRS has the advantage of providing simple diagnostics, a property that IMH does not share.
>
> Concerning the ``significant change'' that you requested, we added in Appendix C.2 of the submission a comparison of the behavior of QRS with both IMH and RWMH for the Poisson experiment, where IMH uses the same global proposal as QRS, and where RWMH uses a local proposal that samples the two neighbours $x-1$ and $x+1$ of the current sample $x$. This comparison shows a clear superiority of QRS and IMH over RWMH and a moderate superiority of QRS over IMH.

---

> > ### Comment · Reviewer_ZGKb · 2022-11-08
> > **Thanks for the responses**
> >
> > Thanks to the authors for their responses to the reviewers. My own concerns have been sufficiently addressed in the responses, especially the inclusion of the IMH and RWMH baselines for the Poisson experiment. The main concerns raise by other reviewers are the relations between this work and gradient based MCMC, and the general claim that large pretrained language models can provide good proposal distributions. For the first concern, the authors provide relevant references of recent gradient-based MCMC samplers in discrete spaces and note that their method bypasses several significant complications that would arise from performing their experiments with gradient-based approaches. This seems reasonable to me. Regarding the quality of pretrained language models as a global proposal distribution, the experimental results are enough to convince me that the proposed method can adjust the global proposal distribution to match constraints in a way that is intuitive to a human, although deeper questions about the quality of the proposal are very difficult to address. Overall, I recommend this paper be accepted.

---

### Review · Reviewer_paAR · 2022-10-11

**Summary Of Contributions:**

The authors propose to study the quasi-rejection sampling to generate samples from discrete data. The main advantage is that the authors can compare the general f-divergence between the sample distribution and target distribution, which provides estimates about how good the samples are compared to the target distribution. Several examples are provided. The proposed algorithm seems to work well in NLP and does over perform existing algorithms, like the well-known  Metropolis-Hastings algorithm. The analysis of the convergence and the estimate of the desired  quantile seems to be the new theoretical contribution.

**Requested Changes:**

The main questions is about the experiment for QRS. Can the authors compare QRS with more recent algorithms, and show that it is the state-of-art in the discrete EBM. Is it possible to compare QRS with continuous model algorithms that haven been mentioned in the discussion.

**Strengths And Weaknesses:**

Strength:

The authors emphasize a lot on the f-divergence which is new for discrete energy based models. The idea is clearly presented and the convergence analysis is established in the appendix.

Weakness:

It seems that Metropolis-Hastings is an relative old algorithm, which is used as the bench mark in the experiments. The authors did mention about the continuous model and algorithms, which seems to not imply application in NLP. My only question here is the following: does the exiting continuous model work for the NLP with some adjustment in the experiment?  (The author said: "none of such prior work has so far  directly addressed NLP applications"). If the continuous model does work, could the author compare with those to show the powerfulness of the QRS? From my understanding, the divergence and convergence of continuous model, e.g. Langevin based algorithms, is quite powerful and well-known. Can the authors compare more why the f-divergence is new and the main difficulty compared to the continuous model here?

---

> ### Author Response · Authors · 2022-10-21
> **Response to Reviewer paAR**
>
> Thank you for your kind words about our work.
>
> Your main request is that we provide details about the relationship of QRS to approaches that attempt to apply advanced gradient-based sampling techniques for continuous models to discrete spaces, in particular in the context of NLP.
>
> In the current related work section of our submission, we provide a brief discussion of certain such approaches, which typically attempt to overcome the lack of a gradient $\nabla_x \log P(x)$, when $x$ belongs to a discrete space $X$, by embedding $X$ in a continuous space $R^d$ and extending $P$ to a differentiable function over that space,
> which is then used to define effective local proposals inside some MH-type random walk sampler. One influential such approach is the recent Gibbs-with-Gradients (GwG) technique [1], which avoids the costly computation of $P(x')$ over all the discrete neighbors $x'$ of a given $x$ by approximating $P(x')$ using a first-order Taylor expansion around $x$.
>
> We will update the related work section to include two more recent references, namely [2, 3] that  adapt a Langevin-style Metropolis Hastings sampler to discrete spaces, and permit a more efficient use of the gradient than GwG, in particular allowing the proposal to update several dimensions at a time, in contrast to GwG.
>
> Concerning relations to NLP, it should be noted that [3], of which we were not aware at the time of our initial submission, does include one NLP application to "text infilling", the ability to replace some blanks in a given sentence by actual words.
>
> However it seems to us that trying to apply any of these gradient-based techniques to the kind of controlled text generation setups that our paper focusses on would involve a major effort beyond the scope of our paper, with difficulties along the following axes:
>
> 1) The gradient-based techniques described above require the sample space $X$ to have a structure naturally organized around $d$ dimensions (this is true for instance of the Ising models, or of the in-filling application), which would need some adaptation for handling whole sentences of varying lengths, as is our case, although we could presumably map each word into some standard embedding.
>
> 2) Several of the EBMs that we consider in our experiments are of the form $P(x) = a(x) b(x)$, where $a(x)$ is a standard autoregressive model, but $b(x)$ can incorporate computation of arbitrary real-valued or binary-valued functions of $x$. Some of these situations --- e.g., to take an extreme example, the case studied in [4], where $x$ is a sequence representing some program code, $a(x)$ is an autoregressive model generating such sequences, and $b(x)$ is a pointwise constraint checking the compilability of $x$ --- would not easily (or at all) lead to a differentiable $P(x)$, even after $x$ had been embedded into some Euclidean structure.
>
> Independently of these issues, the gradient-based approaches cited above do all rely on modifications of Metropolis-Hastings samplers, and presumably inherit their difficulties in evaluating the probabilities of the generated samples. As we have argued, this makes it difficult to estimate the divergence of these samplers from the target EBM, a core advantage of QRS.
>
> [1] Grathwohl et al. 2021 - Oops I took a gradient: Scalable sampling for discrete distributions.
>
> [2] Rhodes and Gutmann 2022 - Enhanced gradient-based MCMC in discrete spaces.
>
> [3] Zhang et al. 2022 - A Langevin-like sampler for discrete distributions
>
> [4] Korbak et al. 2021 - Energy-based models for code generation under compilability constraints

---

### Author Response · Authors · 2022-10-21
**General response to reviewers**

We thank all the reviewers for their thorough and constructive comments and we provide answers and clarifications to each individual reviewer in turn.

We also provide an update to our submission, which may evolve depending on interactions with the reviewers. The point of this provisional update is that it allows us to add some technical material in the main text and in the appendix of the submission that helps support our responses, but which would be inconvenient to include directly in text of the rebuttal. Note: our additions and changes are highlighted in green in the updated submission.

---

### Decision · Action_Editors · 2022-11-14

**Recommendation:** Accept with minor revision

**Comment:**

All three reviewers were satisfied with the revision and are in favour of accepting the paper. Congratulations.

For the camera-ready version, I would like the authors to consider the following two minor requests:

1. Algorithm 1 may need updating as the counter "num_samples_generated" is not incremented in the loop and the hyperparameter num_"samples_target" is likely missing in the list on the first line.

2. In Sec 3.2, I would suggest that you include a brief sentence on the connection to maximum entropy models. You currently refer to the work of Khalifa et al, 2021, which is certainly the main reference, but given the similarity to maximum entropy models, a brief sentence on this connection is likely useful to the reader.


**Audience:**

MCMC is of core interest to TMLR's audience.

**Claims And Evidence:**

The claims made in the paper are supported by theory and simulation results.